# On Divergence Measures for Bayesian Pseudocoresets

**Balhae Kim[1], Jungwon Choi[1], Seanie Lee[1], Yoonho Lee[2], Jung-Woo Ha[3], Juho Lee[1,4]**
KAIST[1], Stanford University[2], NAVER AI Lab[3], AITRICS[4]
{balhaekim, jungwon.choi, lsnfamily02}@kaist.ac.kr,
yoonho@stanford.edu, jungwoo.ha@navercorp.com, juholee@kaist.ac.kr

## Abstract

A Bayesian pseudocoreset is a small synthetic dataset for which the posterior over parameters approximates that of the original dataset. While promising, the scalability of Bayesian pseudocoresets is not yet validated in realistic problems such as image classification with deep neural networks. On the other hand, dataset distillation methods similarly construct a small dataset such that the optimization using the synthetic dataset converges to a solution with performance competitive with optimization using full data. Although dataset distillation has been empirically verified in large-scale settings, the framework is restricted to point estimates, and their adaptation to Bayesian inference has not been explored. This paper casts two representative dataset distillation algorithms as approximations to methods for constructing pseudocoresets by minimizing specific divergence measures: reverse KL divergence and Wasserstein distance. Furthermore, we provide a unifying view of such divergence measures in Bayesian pseudocoreset construction. Finally, we propose a novel Bayesian pseudocoreset algorithm based on minimizing forward KL divergence. Our empirical results demonstrate that the pseudocoresets constructed from these methods reflect the true posterior even in high-dimensional Bayesian inference problems.

## 1 Introduction

One of the main ingredients for modern statistical machine learning is large-scale data, which enables the learning of powerful and flexible models such as deep neural networks when handled correctly [3, 32]. However, training a neural network on larger datasets usually requires many nontrivial choices, ranging from architecture choice, hyperparameter optimization, and infrastructure challenges [33]. Aside from requiring enormous computing resources for training, large-scale data often contains sensitive information that must not be shared publicly [11]. This motivates the construction of *coresets* [29, 14], a sparse subset of a large-scale dataset that summarizes essential features of the original dataset. In the context of Bayesian inference, the posterior conditioned on an essential coreset should be similar to the exact posterior conditioned on the full dataset.

Coreset construction becomes more difficult with high-dimensional data. A previous work [19] shows that even the best coreset becomes suboptimal at large scales: the divergence between the posterior conditioned on the optimal coreset and the exact posterior grows with the dimension of the data. Motivated by this weakness of strict coresets, Manousakas et al. [19] proposes to construct a *Bayesian pseudocoreset*: a synthetic dataset represented as a set of learnable parameters and trained to minimize the divergence between the coreset posterior and the full-data posterior. Compared to coresets, pseudocoresets scale much better with data dimension and achieve significantly lower posterior approximation error. However, the KL divergence minimization in pseudocoreset learning requires the construction of approximate posteriors on the fly during the learning process, making this approach hard to apply to high-dimensional models such as deep neural networks.

36th Conference on Neural Information Processing Systems (NeurIPS 2022).

On the other hand, *dataset distillation* [28] aims to distill a large-scale dataset into a small synthetic dataset such that the test performance of a model trained on the distilled data is comparable to that of a model trained on the original data. Such methods have achieved remarkable performance in real-world problems with high-dimensional data and deep neural networks. Although dataset distillation methods have a similar motivation to pseudocoresets, the two methods optimize different objectives. Whereas pseudocoresets minimize the divergence between coreset and full-data posteriors, dataset distillation considers a non-Bayesian setting with heuristically set objective functions such as matching the gradients of loss functions [36] or matching the parameters obtained from optimization trajectories [8]. Such objectives have no direct Bayesian analogue due to their heuristic nature, and more theoretical grounding can advance our understanding of dataset distillation and the empirical performance of Bayesian pseudocoreset methods.

In this paper, we provide a unifying view of Bayesian pseudocoresets and dataset distillation to take advantage of both approaches. We first study various choices of divergence measures as objectives for learning pseudocoresets. While Manousakas et al. [19] minimizes the reverse KL divergence between the pseudocoreset posterior distribution and the full data posterior distribution, we show that alternative divergence measures such as forward KL divergence and Wasserstein distance are also effective in practice. Based on this perspective, we re-interpret existing dataset distillation algorithms [36, 8] approximations to the Bayesian pseudocoresets learned by minimizing reverse KL and Wasserstein distance, with specific choices of variational approximations for coreset posteriors. This connection justifies the heuristically chosen learning objectives of dataset distillation algorithms, and at the same time, provides a theoretical background for using the distilled datasets obtained from the such procedure for Bayesian inference. Conversely, Bayesian pseudocoresets benefit from this connection by borrowing various ideas and tricks used in the dataset distillation algorithms to make them scale for complex tasks. For instance, the variational posteriors we identified by recasting dataset distillation algorithms as Bayesian pseudocoreset learning can be used for Bayesian pseudocoresets with various choices of divergence measures. Also, we can adapt the idea of reusing pre-computed optimization trajectories [8] which have already been shown to work at large scales.

We empirically compare pseudocoreset algorithms based on three different divergence measures on high-dimensional image classification tasks. Our results demonstrate that we can efficiently and scalably construct Bayesian pseudocoresets with which MCMC algorithms such as Hamiltonian Monte-Carlo (HMC) [10, 22] or Stochastic Gradient Hamiltonian Monte-Carlo (SGHMC) [9] accurately approximate the full posterior.

## 2 Background

### 2.1 Bayesian Pseudocoresets

Denote observed data as $\mathbf{x} = \{x_n\}_{n=1}^N$. Given a probabilistic model indexed by a parameter $\theta$ with some prior distribution $\pi_0$, we are interested in the posterior conditioned on the full data $\mathbf{x}$,

$$\pi_{\mathbf{x}}(\theta) = \frac{1}{Z(\mathbf{x})} \exp\left(\sum_{n=1}^N f(x_n, \theta)\right) \pi_0(\theta) \coloneqq \frac{1}{Z(\mathbf{x})} \exp\left(\mathbb{1}_N^\top \mathbf{f}(\mathbf{x}, \theta)\right) \pi_0(\theta), \tag{1}$$

where $\mathbf{f}(\mathbf{x}, \theta) \coloneqq [f(x_1, \theta), \ldots, f(x_N, \theta)]^\top$, $\mathbb{1}_N$ is the $N$-dimensional one vector, and $Z(\mathbf{x}) = \int \exp(\mathbb{1}_N^\top \mathbf{f}(\mathbf{x}, \theta)) \mathrm{d}\theta$ is a partition function. The posterior $\pi_{\mathbf{x}}$ is usually intractable to compute due to $Z(\mathbf{x})$, so we employ approximate inference algorithms. Bayesian pseudocoreset methods [19] aim to construct a synthetic dataset called a *pseudocoreset* $\mathbf{u} = \{u_m\}_{m=1}^M$ with $M \ll N$ such that the posterior of $\theta$ conditioned on it approximates the original posterior $\pi_{\mathbf{x}}$. The pseudocoreset posterior is written as[1],

$$\pi_{\mathbf{u}}(\theta) = \frac{1}{Z(\mathbf{u})} \exp\left(\sum_{m=1}^M f(u_m, \theta)\right) \pi_0(\theta) = \frac{1}{Z(\mathbf{u})} \exp(\mathbb{1}_M^\top \mathbf{f}(\mathbf{u}, \theta)) \pi_0(\theta), \tag{2}$$

---

[1]Manousakas et al. [19] considered two sets of parameters, the coreset elements $\mathbf{u}$ and their weights $\mathbf{w} = \{w_m\}_{m=1}^M$. We empirically found that the weights $\mathbf{w}$ have a negligible impact on performance, so we only learn the coreset elements.

where $\mathbf{f}(\mathbf{u}, \theta)$, $\mathbb{1}_M$, and $Z(\mathbf{u})$ are defined similarly. Manousakas et al. [19] suggests to learn $\mathbf{u}$ by minimizing the KL divergence between $\pi_\mathbf{u}$ and $\pi_\mathbf{x}$:

$$\mathbf{u}^* = \arg\min_{\mathbf{u}} \ D_{\mathrm{KL}}[\pi_\mathbf{u} \| \pi_\mathbf{x}], \tag{3}$$

and show that the gradient of this objective is computed as

$$\nabla_{u_m} D_{\mathrm{KL}}[\pi_\mathbf{u} \| \pi_\mathbf{x}] = -\mathrm{Cov}_{\pi_\mathbf{u}(\theta)} \left[ \nabla_u \mathbf{f}(u_m, \theta), \mathbb{1}_N^\top \mathbf{f}(\mathbf{x}, \theta) - \mathbb{1}_M^\top \mathbf{f}(\mathbf{u}, \theta) \right]. \tag{4}$$

The expectation over the pseudocoreset posterior $\pi_\mathbf{u}$ in the above equation is further approximated with the Monte-Carlo estimation with an approximate posterior $q_\mathbf{u} \approx \pi_\mathbf{u}$, for instance, from the Laplace approximation.

## 2.2 Dataset Distillation

With a similar motivation as Bayesian pseudocoresets, dataset distillation methods construct a small synthetic dataset $\mathbf{u}$, but from the perspective of matching optimal parameters. That is, in dataset distillation, we find $\mathbf{u}$ such that

$$\mathbf{u}^* = \arg\min_{\mathbf{u}} \ d(\theta_\mathbf{x}^*, \theta_\mathbf{u}^*), \quad \theta_\mathbf{x}^* = \arg\min_{\theta} \ \ell(\mathbf{x}, \theta), \quad \theta_\mathbf{u}^* = \arg\min_{\theta} \ \ell(\mathbf{u}, \theta), \tag{5}$$

with a loss function $\ell(\cdot, \theta)$ and a distance function $d(\cdot, \cdot)$ comparing two parameters. In general, optimizing $d(\theta_\mathbf{x}^*, \theta_\mathbf{u}^*)$ is intractable since it is a bilevel optimization problem requiring $\theta_\mathbf{x}^*$ and $\theta_\mathbf{u}^*$, so existing works relax the problem in several ways, such as using kernel-based approaches [23, 24] or solving only for a short horizon in inner optimization [36, 34, 8]. Below, we briefly review two representative methods—Dataset Condensation (DC) [36] and Matching Training Trajectories (MTT) [8].

**Dataset Condensation**   Instead of fully unrolling the inner optimization process, DC instead computes the matching objective at *every inner step* starting from some random parameter $\theta^{(0)}$.

$$\mathbf{u}^* = \arg\min_{\mathbf{u}} \ \mathbb{E}_{\theta^{(0)}} \left[ \sum_{t=0}^{T} d_{\cos} \left( \nabla_\theta \ell(\mathbf{x}, \theta^{(t)}), \nabla_\theta \ell(\mathbf{u}, \theta^{(t)}) \right) \right], \tag{6}$$

$$\theta^{(t)} = \theta^{(t-1)} - \eta \nabla_\theta \ell(\mathbf{u}, \theta^{(t-1)}) \text{ for } t \geq 1, \tag{7}$$

where $d_{\cos}$ denotes the cosine similarity between two gradients. To implement this objective, we build a trajectory of parameters starting from a random $\theta^{(0)}$ with the current distilled dataset $\mathbf{u}$. We then compute the gradient matching objective for each layer at *each step* of the trajectory to update $\mathbf{u}$. Despite its appealing simplicity, this algorithm may suffer from short-horizon bias due to the limited trajectory length.

**Matching Training Trajectories**   MTT extends the single-step gradient matching in DC and considers longer inner optimization trajectories. Starting from a randomly initialized parameter $\theta^{(0)}$, MTT takes multiple inner optimization steps with both $\mathbf{u}$ and $\mathbf{x}$, and minimizes the normalized $L_2$ distance between two parameters at the end of the learning trajectories.

$$\mathbf{u}^* = \arg\min_{\mathbf{u}} \ \mathbb{E}_{\theta^{(0)}} \left[ d_2 \left( \theta_\mathbf{x}^{(L_\mathbf{x})}, \theta_\mathbf{u}^{(L_\mathbf{u})} \right) \right], \quad \theta_\mathbf{x}^{(0)} = \theta_\mathbf{u}^{(0)} = \theta^{(0)}, \tag{8}$$

$$\theta_\mathbf{x}^{(t)} = \theta_\mathbf{x}^{(t-1)} - \eta_\mathbf{x} \nabla_\theta \ell(\mathbf{x}, \theta_\mathbf{x}^{(t-1)}) \text{ for } t = 1, \dots, L_\mathbf{x}, \tag{9}$$

$$\theta_\mathbf{u}^{(t)} = \theta_\mathbf{u}^{(t-1)} - \eta_\mathbf{u} \nabla_\theta \ell(\mathbf{u}, \theta_\mathbf{u}^{(t-1)}) \text{ for } t = 1, \dots, L_\mathbf{u}, \tag{10}$$

where $d_2(\cdot, \cdot)$ is the normalized $L_2$ distance between parameters. Unlike DC, this algorithm starts from a common initial parameter $\theta^{(0)}$, keeps track of two separate optimization trajectories (one with $\mathbf{u}$ and another with $\mathbf{x}$), and the outer-update for $\mathbf{u}$ is done only after $L_\mathbf{x}$ and $L_\mathbf{u}$ inner steps. To reduce the heavy computational cost for the inner update with the full dataset $\mathbf{x}$, MTT employs mini-batch SGD for the inner steps, in addition to saving and re-using precomputed SGD trajectories to train $\mathbf{u}$. The use of precomputed SGD trajectories is similar to that of replay buffers for reinforcement learning [17, 20, 1], and allows MTT to handle long inner-loop sequences. While MTT significantly improves over DC, presumably due to its longer inner optimization horizon, the algorithm requires large memory for even a moderate number of inner-steps $L_\mathbf{u}$ because it requires backpropagation through all inner steps with $\mathbf{u}$.

# 3 On Divergence Measures for Bayesian Pseudocoresets

While the Bayesian pseudocoreset is originally obtained by minimizing the reverse KL divergence, in principle we can minimize an arbitrary divergence measure $D(\cdot, \cdot)$ to achieve the goal:

$$\mathbf{u}^* = \arg\min_{\mathbf{u}} \ D(\pi_{\mathbf{x}}, \pi_{\mathbf{u}}). \tag{11}$$

In this section, we study three different choices for such divergence measures along with their relation to dataset distillation methods.

## 3.1 Reverse KL Divergence

Manousakas et al. [19] proposed to minimize reverse KL divergence with a Laplace approximation to obtain the pseudocoreset posterior $\pi_{\mathbf{u}}$. Here, we show that it is possible to recast Dataset Condensation (DC) as a reverse KL divergence minimization with an alternative choice for the variational posterior. Let $\theta^{(t)}$ be a parameter in a learning trajectory with a loss function $\ell(\mathbf{u}, \theta) := -\mathbb{1}_M^\top \mathbf{f}(\mathbf{u}, \theta)$. Provided that $\theta^{(t)}$ is sufficiently close to a local optimum, we can construct a naïve approximation of $\pi_{\mathbf{u}}$ as

$$q_{\mathbf{u}}(\theta) = \mathcal{N}(\theta; \theta^{(t)}, \Sigma), \quad \theta^{(t)} = \theta^{(t-1)} - \eta \nabla_\theta \ell(\mathbf{u}, \theta^{(t-1)}). \tag{12}$$

Under this setting, we can actually show that one gradient descent step with respect to reverse KL divergence $D_{\mathrm{KL}}[\pi_{\mathbf{u}} \| \pi_{\mathbf{x}}]$ is equivalent to one step of gradient matching in DC when the parameter being optimized is close to local optima. We note that while the Gaussian posterior is a simple approximation class, it is commonly used due to its demonstrated usefulness in several literatures [18], such as the Bernstein-von Mises theorem [16, 27]. However, we also note that these Gaussian approximations are not necessary for establishing the connection between the dataset distillation and Bayesian pseudocoresets; Gaussian approximations are just a special case which has the benefit of simplifying the analysis.

**Proposition 3.1.** *Let $q_{\mathbf{u}}(\theta)$ set as Eq. 12. Assume $\theta^{(t-1)}$ is close enough to local optima, so that we have $\|\theta^{(t-1)} - \theta^{(t)}\| \ll 1$. Then we have*

$$\nabla_{\mathbf{u}} D_{\mathrm{KL}}[\pi_{\mathbf{u}} \| \pi_{\mathbf{x}}] \approx -\eta \nabla_{\mathbf{u}} \Big( \nabla_\theta \ell(\mathbf{x}, \theta^{(t-1)})^\top \nabla_\theta \ell(\mathbf{u}, \theta^{(t-1)}) \Big). \tag{13}$$

The proof is given in Appendix A. In other words, when sufficiently close to local optima, DC can be interpreted as a special instance of Bayesian pseudocoreset via reverse KL minimization with Gaussian approximation for the pseudocoreset posterior $\pi_{\mathbf{u}}$. Throughout the paper, we will refer to the Bayesian pseudocoreset with reverse KL minimization as BPC-rKL. To simply implement the Algorithm 1 in [19] for the image dataset, we approximated $\pi_{\mathbf{u}}$ as a Gaussian distribution. Please refer to Appendix B.2 for the detailed algorithm.

## 3.2 Wasserstein Distance

We can instead choose $D(\cdot, \cdot)$ to be the Wasserstein distance between $\pi_{\mathbf{u}}$ and $\pi_{\mathbf{x}}$. While the Wasserstein distance is intractable in general, by approximating both $\pi_{\mathbf{u}}$ and $\pi_{\mathbf{x}}$ with Gaussian distributions, we can attain a closed-form expression for this distance metric. Specifically, let $\theta_{\mathbf{u}}$ and $\theta_{\mathbf{x}}$ be two parameters sufficiently close to local optima. Then we can construct a crude approximation for $\pi_{\mathbf{u}}$ and $\pi_{\mathbf{x}}$ as,

$$\pi_{\mathbf{u}}(\theta) \approx q_{\mathbf{u}}(\theta) := \mathcal{N}(\theta; \theta_{\mathbf{u}}, \Sigma_{\mathbf{u}}), \quad \pi_{\mathbf{x}}(\theta) \approx q_{\mathbf{x}}(\theta) := \mathcal{N}(\theta; \theta_{\mathbf{x}}, \Sigma_{\mathbf{x}}). \tag{14}$$

Then the Wasserstein distance between these two distributions is computed as

$$W_2(q_{\mathbf{u}}; q_{\mathbf{x}}) = \|\theta_{\mathbf{u}} - \theta_{\mathbf{x}}\|_2^2 + \mathrm{Tr}\Big( \Sigma_{\mathbf{x}} + \Sigma_{\mathbf{u}} - 2(\Sigma_{\mathbf{u}}^{1/2} \Sigma_{\mathbf{x}} \Sigma_{\mathbf{u}}^{1/2})^{1/2} \Big). \tag{15}$$

Now consider a single outer-step of the MTT. Starting from a common initial $\theta_0$, the algorithm takes the gradient steps both with $\mathbf{x}$ and $\mathbf{u}$ to get $\theta_{\mathbf{x}}^{(L_{\mathbf{x}})}$ and $\theta_{\mathbf{u}}^{(L_{\mathbf{u}})}$. If we set $q_{\mathbf{u}}(\theta) = \mathcal{N}(\theta; \theta_{\mathbf{u}}^{(L_{\mathbf{u}})}, \Sigma)$ and $q_{\mathbf{x}}(\theta) = \mathcal{N}(\theta; \theta_{\mathbf{x}}^{(L_{\mathbf{x}})}, \Sigma)$ with a common covariance $\Sigma$, then the Wasserstein distance reduces to $\|\theta_{\mathbf{x}}^{(L_{\mathbf{x}})} - \theta_{\mathbf{u}}^{(L_{\mathbf{u}})}\|_2^2$, which is proportional to the $L_2$ objective being optimized in the MTT. Similarly to the reverse KL divergence and BPC-rKL, we refer to the Bayesian pseudocoreset with Wasserstein distance minimization as BPC-W.

### 3.3 Forward KL Divergence

Finally, we consider an alternative Bayesian pseudocoreset algorithm based on minimizing forward KL divergence. The forward KL minimization is known to encourage a model to cover the entire target distribution while the reverse KL minimization favors mode capturing models, indicating that the forward KL may be a better choice for the Bayesian pseudocoreset. Under our setting, the forward KL is computed as

$$D_{\mathrm{KL}}[\pi_{\mathbf{x}} \| \pi_{\mathbf{u}}] = \log Z(\mathbf{u}) - \log Z(\mathbf{x}) + \mathbb{E}_{\pi_{\mathbf{x}}}[\mathbb{1}_N^\top \mathbf{f}(\mathbf{x}, \theta)] - \mathbb{E}_{\pi_{\mathbf{x}}}[\mathbb{1}_M^\top \mathbf{f}(\mathbf{u}, \theta)]. \tag{16}$$

Then the gradient w.r.t. the pseudocoreset $\mathbf{u}$ is computed as,

$$\begin{aligned}
\nabla_{\mathbf{u}} D_{\mathrm{KL}}[\pi_{\mathbf{x}} \| \pi_{\mathbf{u}}] &= \nabla_{\mathbf{u}} \log Z(\mathbf{u}) - \nabla_{\mathbf{u}} \mathbb{E}_{\pi_{\mathbf{x}}}[\mathbb{1}_M^\top \mathbf{f}(\mathbf{u}, \theta)] \\
&= \mathbb{E}_{\pi_{\mathbf{u}}}[\nabla_{\mathbf{u}}(\mathbb{1}_M^\top \mathbf{f}(\mathbf{u}, \theta))] - \nabla_{\mathbf{u}} \mathbb{E}_{\pi_{\mathbf{x}}}[\mathbb{1}_M^\top \mathbf{f}(\mathbf{u}, \theta)].
\end{aligned} \tag{17}$$

Again, we should approximate the expectation w.r.t. $\pi_{\mathbf{u}}$ and $\pi_{\mathbf{x}}$ to obtain this gradient. We introduce similar variational posteriors,

$$q_{\mathbf{u}}(\theta) = \mathcal{N}(\theta; \theta_{\mathbf{u}}^{(L_{\mathbf{u}})}, \Sigma_{\mathbf{u}}), \quad q_{\mathbf{x}}(\theta) = \mathcal{N}(\theta; \theta_{\mathbf{x}}^{(L_{\mathbf{x}})}, \Sigma_{\mathbf{x}}). \tag{18}$$

The endpoint $\theta_{\mathbf{u}}^{(L_{\mathbf{u}})}$ is a function of $\mathbf{u}$, so in principle, the expectation w.r.t. $q_{\mathbf{u}}$ involves unrolling through the parameter trajectories $\theta^{(0)} \to \theta_{\mathbf{u}}^{(1)} \to \cdots \to \theta_{\mathbf{u}}^{(L_{\mathbf{u}})}$, as for the BPC-W. We employ truncated backpropagation to reduce memory cost due to this unrolling, i.e., we block the gradient flow through $\theta^{(L_{\mathbf{u}})}$ to obtain

$$\begin{aligned}
\nabla_{\mathbf{u}} D_{\mathrm{KL}}[\pi_{\mathbf{x}} \| \pi_{\mathbf{u}}] &\approx \mathbb{E}_{q_{\mathbf{u}}}[\nabla_{\mathbf{u}}(\mathbb{1}_M^\top \mathbf{f}(\mathbf{u}, \theta))] - \nabla_{\mathbf{u}} \mathbb{E}_{q_{\mathbf{x}}}[\mathbb{1}_M^\top \mathbf{f}(\mathbf{u}, \theta)] \\
&= \mathbb{E}_{\varepsilon_{\mathbf{u}}}[\nabla_{\mathbf{u}}(\mathbb{1}_M^\top \mathbf{f}(\mathbf{u}, \mathbf{sg}(\theta_{\mathbf{u}}^{(L_{\mathbf{u}})}) + \Sigma_{\mathbf{u}}^{1/2}\varepsilon_{\mathbf{u}}))] - \nabla_{\mathbf{u}} \mathbb{E}_{\varepsilon_{\mathbf{x}}}[\mathbb{1}_M^\top \mathbf{f}(\mathbf{u}, \theta_{\mathbf{x}}^{(L_{\mathbf{x}})} + \Sigma_{\mathbf{x}}^{1/2}\varepsilon_{\mathbf{x}})] \\
&= \nabla_{\mathbf{u}}\left(\mathbb{E}_{\varepsilon_{\mathbf{u}}}[\mathbb{1}_M^\top \mathbf{f}(\mathbf{u}, \mathbf{sg}(\theta_{\mathbf{u}}^{(L_{\mathbf{u}})}) + \Sigma_{\mathbf{u}}^{1/2}\varepsilon_{\mathbf{u}})] - \mathbb{E}_{\varepsilon_{\mathbf{x}}}[\mathbb{1}_M^\top \mathbf{f}(\mathbf{u}, \theta_{\mathbf{x}}^{(L_{\mathbf{x}})} + \Sigma_{\mathbf{x}}^{1/2}\varepsilon_{\mathbf{x}})]\right),
\end{aligned} \tag{19}$$

where $\mathbf{sg}(\cdot)$ denotes the stop-grad operation. We further approximate this via Monte-Carlo estimation,

$$\approx \nabla_{\mathbf{u}}\left(\frac{1}{S}\sum_{s=1}^{S}\left(\mathbb{1}_M^\top \mathbf{f}(\mathbf{u}, \mathbf{sg}(\theta_{\mathbf{u}}^{(L_{\mathbf{u}})}) + \Sigma_{\mathbf{u}}^{1/2}\varepsilon_{\mathbf{u}}^{(s)}) - \mathbb{1}_M^\top \mathbf{f}(\mathbf{u}, \theta_{\mathbf{x}}^{(L_{\mathbf{x}})} + \Sigma_{\mathbf{x}}^{1/2}\varepsilon_{\mathbf{x}}^{(s)})\right)\right), \tag{20}$$

with $\varepsilon_{\mathbf{x}}^{(1)}, \ldots, \varepsilon_{\mathbf{x}}^{(S)}, \varepsilon_{\mathbf{u}}^{(1)}, \ldots, \varepsilon_{\mathbf{u}}^{(S)} \overset{\text{i.i.d.}}{\sim} \mathcal{N}(0, I)$. The resulting algorithm starts from a common initial parameter $\theta^{(0)}$, keeps track of two parameter trajectories, and minimizes the difference in the log-likelihood $\mathbb{1}_M^\top \mathbf{f}(\mathbf{u}, \theta)$ evaluated at the Gaussian neighbors of the endpoints of the two trajectories. Note that even though we block gradient flow through $\theta_{\mathbf{u}}^{(L_{\mathbf{u}})}$, the pseudocoreset $\mathbf{u}$ still influences each likelihood term $\mathbf{f}(\mathbf{u}, \theta)$, so the truncated gradient includes meaningful learning signals. In contrast, if we apply a similar trick to the BPC-W, the matching objective would be $\|\mathbf{sg}(\theta_{\mathbf{u}}^{(L_{\mathbf{u}})}) - \theta_{\mathbf{x}}^{(L_{\mathbf{x}})}\|$, which is constant w.r.t. $\mathbf{u}$. We can additionally borrow an idea from MTT, where we utilize the set of expert trajectories to efficiently evaluate the parameter trajectory with $\mathbf{x}$. We call this overall algorithm BPC-fKL, and provide a summary of the overall procedure in Algorithm 1.

**Relation to contrastive divergence**  BPC-fKL is closely related to contrastive divergence [13, 7], a learning algorithm for approximating the maximum likelihood estimator. Suppose we want to train a model $p_\theta(x) = \frac{1}{Z(\theta)}\exp(f_\theta(x))$ by maximizing the log-likelihood function of training examples $\{x_n\}_{n=1}^N \sim p_{\text{data}}(x)$,

$$L(\theta) = \frac{1}{N}\sum_{n=1}^{N}\log p_\theta(x_n). \tag{21}$$

Then the gradient of the log-likelihood is

$$L'(\theta) = \frac{1}{N}\sum_{n=1}^{N}\nabla_\theta f_\theta(x_n) - \mathbb{E}_{p_\theta(x)}[\nabla_\theta f_\theta(x)] = \mathbb{E}_{p_{\text{data}}(x)}[\nabla_\theta f_\theta(x)] - \mathbb{E}_{p_\theta(x)}[\nabla_\theta f_\theta(x)]. \tag{22}$$

Contrastive divergence approximates this gradient by approximating the second term with sampling methods such as Langevin dynamics or HMC. By doing this we can obtain the MLE for log-likelihood

---

**Algorithm 1** Bayesian Pseudocoresets with Forward KL

---

**Require:** Set of expert parameter trajectories $\{\tau\}$ trained with $\mathbf{x}$, each parameter trajectory saves parameters at the end of training epochs.

**Require:** Number of updates with the pseudocoreset (full data) $L_{\mathbf{u}}(L_{\mathbf{x}})$, total training steps $K$, maximum start epoch $T^+$, number of Gaussian samples $S$, variances $\Sigma_{\mathbf{u}}, \Sigma_{\mathbf{x}}$, learning rate $\eta$.

**Require:** Differentiable augmentation function $\mathcal{A}$ (Optional).

Initialize the pseudocoreset $\mathbf{u}$ by randomly selecting a subset of size $M$ from $\mathbf{x}$.

**for** $k = 1, \dots, K$ **do**

  Sample an expert trajectory $\tau = \{\theta_*^{(r)}\}_{r=0}^T$.

  Randomly choose an epoch to start $r \le T^+$ and initialize $\theta_{\mathbf{u}}^{(0)} = \theta_{\mathbf{x}}^{(0)} = \theta_*^{(r)}$.

  Obtain $\theta_{\mathbf{x}}^{(L_{\mathbf{x}})} = \theta_*^{(r+L_{\mathbf{x}})}$.

  **for** $t = 1, \dots, L_{\mathbf{u}}$ **do**

    Update the network parameter $\theta_{\mathbf{u}}^{(t)} \leftarrow \theta_{\mathbf{u}}^{(t-1)} + \eta \nabla \mathbb{1}_M^\top \mathbf{f}(\mathcal{A}(\mathbf{u}), \theta_{\mathbf{u}}^{(t-1)})$.

  **end for**

  Sample random Gaussian noises $\{\varepsilon_{\mathbf{x}}^{(s)}, \varepsilon_{\mathbf{u}}^{(s)}\}_{s=1}^S \overset{\text{i.i.d.}}{\sim} \mathcal{N}(0, I)$.

  Update the pseudocoreset with the gradient using Eq. 20.

**end for**

---

without knowing the exact potential function. Our proposed BPC-fKL performs the same function as the contrastive divergence if the model parameter $\theta$ and training data $\mathbf{x}$ change to the Bayesian pseudocoresets $\mathbf{u}$ and the parameters from the true posterior distribution which are approximated by the points near the expert trajectories. In this point of view, BPC-fKL is the maximum likelihood estimator that maximizes the log-likelihood of the parameters from the posterior distribution of the original dataset.

# 4 Related Works

**Bayesian coresets**    As running algorithms such as MCMC and VI on large datasets is challenging due to expensive computational cost, Bayesian coresets [14], methods that represent the entire dataset as a sparse and weighted sum of subsets, have been studied. Campbell and Broderick [6] interprets coreset construction as a sparse approximation of vector sums and generalizes it to a sparse regression problem in a Hilbert space. Campbell and Broderick [5] demonstrates that Hilbert coresets scale the coresets log-likelihood sub-optimally, and presents a modified algorithm. As Hilbert coresets include the choice of weighted $L^2$ inner product or finite-dimensional projection and it is difficult to choose them optimally, another method that minimizes the KL divergence between the coreset posterior and true posterior has also been studied recently [4]. Bayesian coresets have the advantages of simple and theoretically guaranteed quality of the posterior approximations. However, for high-dimensional data, considering only subsets of the dataset fails as even an optimal coreset has a KL divergence that increases with data dimension. Additionally, privacy concerns make subset-based coresets difficult to apply to real-world conditions. Bayesian pseudocoresets [19], constructed from learned synthetic datapoints, can avoid these shortcomings of coresets, but have been empirically validated only in relatively easier low-dimensional settings such as logistic regression. To our best knowledge, this paper is the first to experimentally demonstrate the viability of Bayesian pseudocoresets for high-dimensional real data.

**Dataset distillation**    The goal of dataset distillation [28] is to create a small synthetic dataset that allows the model to have similar test performance to the original dataset even after training with the smaller synthetic dataset. Such synthetic datasets have potential applications in many subfields of machine learning such as continual learning and neural architecture search. However, dataset distillation typically involves a bilevel optimization structure in the learning process, which suffers from high computational costs or training instability. Existing methods alleviate these challenges by matching one-step gradients between synthetic and real data [36, 34], solving using a closed form solution of kernel ridge regression [23, 24], or reducing the normalized distance between parameters at the end of synthetic and real training trajectories [8]. However, because these methods focus only on high test accuracy, they do not consider uncertainty or the degree to which the posterior distribution on the synthetic dataset matches the true posterior distribution. While Zhao and Bilen [35] matches the distributions of embedding features for many random networks of synthetic and real data, this

**Table 1:** Performance of each Bayesian pseudocoreset method with $\{1, 10, 20\}$ images per class (ipc) on the CIFAR10 test dataset. We present results with HMC and A-SGHMC. The SGHMC results for the entire dataset are $0.7383 \pm 0.0052$ accuracy and $0.9387 \pm 0.0152$ nll. Aug denotes image augmentation during training. All values are averaged over ten random seeds.

| | | HMC | | HMC (+Aug) | | A-SGHMC (+Aug) | |
|---|---|---|---|---|---|---|---|
| | | Acc ($\uparrow$) | NLL ($\downarrow$) | Acc ($\uparrow$) | NLL ($\downarrow$) | Acc ($\uparrow$) | NLL ($\downarrow$) |
| 1 ipc | Random | $0.1745\pm0.0084$ | $2.4507\pm0.0657$ | $0.1745\pm0.0084$ | $2.4507\pm0.0657$ | $0.1414\pm0.0045$ | $2.9798\pm0.0725$ |
| | BPC-rKL | $0.2472\pm0.0120$ | $2.1635\pm0.0327$ | $0.2416\pm0.0133$ | $2.1688\pm0.0486$ | $0.2162\pm0.0083$ | $2.4617\pm0.0841$ |
| | BPC-W (MTT) | $\mathbf{0.3435}\pm0.0207$ | $\mathbf{1.9311}\pm0.0329$ | $0.3338\pm0.0158$ | $1.9589\pm0.0265$ | $\mathbf{0.2934}\pm0.0121$ | $\mathbf{2.1400}\pm0.0333$ |
| | BPC-fKL | $0.3354\pm0.0066$ | $2.0253\pm0.0311$ | $\mathbf{0.3468}\pm0.0119$ | $\mathbf{1.9574}\pm0.0269$ | $0.2823\pm0.0128$ | $2.1426\pm0.0343$ |
| 10 ipc | Random | $0.2807\pm0.0050$ | $2.1070\pm0.0178$ | $0.2807\pm0.0050$ | $2.1070\pm0.0178$ | $0.3085\pm0.0077$ | $2.1953\pm0.0481$ |
| | BPC-rKL | $0.3253\pm0.0075$ | $1.9627\pm0.0312$ | $0.3017\pm0.0074$ | $2.0251\pm0.0134$ | $0.3789\pm0.0154$ | $1.9492\pm0.0605$ |
| | BPC-W (MTT) | $0.3535\pm0.0127$ | $1.9450\pm0.0258$ | $0.3646\pm0.0082$ | $1.9307\pm0.0171$ | $\mathbf{0.4890}\pm0.0172$ | $1.6971\pm0.0392$ |
| | BPC-fKL | $\mathbf{0.4294}\pm0.0101$ | $\mathbf{1.7292}\pm0.0248$ | $\mathbf{0.4252}\pm0.0091$ | $\mathbf{1.7334}\pm0.0217$ | $0.4878\pm0.0103$ | $\mathbf{1.6762}\pm0.0390$ |
| 20 ipc | Random | $0.3292\pm0.0057$ | $2.0164\pm0.0212$ | $0.3292\pm0.0057$ | $2.0164\pm0.0212$ | $0.3832\pm0.0070$ | $1.8999\pm0.0338$ |
| | BPC-rKL | $0.3686\pm0.0115$ | $1.8848\pm0.0338$ | $0.3518\pm0.0102$ | $1.9540\pm0.0350$ | $0.4307\pm0.0073$ | $1.8555\pm0.0263$ |
| | BPC-W (MTT) | $0.4546\pm0.0146$ | $1.7336\pm0.0325$ | $0.4606\pm0.0200$ | $1.7334\pm0.0506$ | $\mathbf{0.5691}\pm0.0167$ | $\mathbf{1.5439}\pm0.0408$ |
| | BPC-fKL | $\mathbf{0.4910}\pm0.0088$ | $\mathbf{1.6279}\pm0.0264$ | $\mathbf{0.4833}\pm0.0069$ | $1.6440\pm0.0231$ | $0.5153\pm0.0093$ | $1.6701\pm0.0256$ |

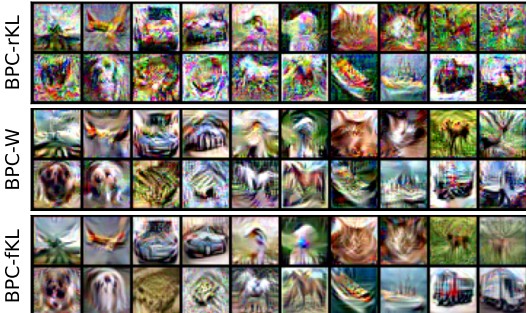

**Figure 1:** Examples of Bayesian pseudocoresets.

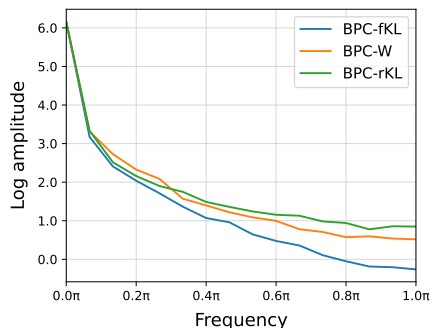

**Figure 2:** Log amplitude in frequency domain.

work similarly does not consider posterior distributions. This paper re-interprets and extends scalable dataset distillation methods to the problem of matching posterior distribution matching.

# 5 Experiments

## 5.1 Experimental Setup

**Datasets and model architecture**    We use the CIFAR10 dataset [15] to generate Bayesian pseudocoresets, and evaluate on the test split of CIFAR10 in addition to the CIFAR10-C dataset [12], which imposes additional uncertainty through image corruptions. Following the experimental setup of previous works [36, 34], we use a 3-layer ConvNet as the network architecture.

**Evaluation**    We obtain pseudocoresets from three Bayesian pseudocoreset construction methods: BPC-fKL, BPC-W, and BPC-rKL. We run two Markov chain Monte Carlo methods (HMC [22] and SGHMC [9]) and report the top-1 accuracy and negative log-likelihood (NLL) with respect to ground-truth labels. Note that pseudocoresets are small enough to run full-batch HMC in our experimental setting, and SGHMC which originally subsamples mini-batches from the full data is not a favorable option in our situation. Instead, at each iteration, while holding the batch (pseudocoreset) fixed, we apply random data augmentation to the batch. As a result, the gradients computed from the randomly augmented batches differ at each iteration, and we run SGHMC with these stochastic gradients. To distinguish this setting from the typical mini-batch SGHMC, we denote this as Altered-SGHMC (A-SGHMC). As our results were not sensitive to the choice of hyperparameters, we used a single set of hyperparameters that performed best in initial experiments. Please refer to Appendix B for detailed evaluation settings including hyperparameters.

## 5.2 Main Results

We experimentally evaluate the effectiveness of the three pseudocoreset construction methods previously discussed. We consider three different settings corresponding to different memory budgets for the pseudocoresets: $n \in \{1, 10, 20\}$ images per class (ipc). In each setting, we consider a random coreset baseline, in which we randomly subsample $n$ images from the original dataset. To further evaluate the effectiveness of data augmentation in pseudocoreset training, we show performance of each coreset method with and without augmentations. We additionally examine the role of augmentations at test time through A-SGHMC.

Results in Table 1 show that all Bayesian pseudocoresets have higher accuracy and lower negative log-likelihood compared to random coresets. BPC-rKL is slightly worse than BPC-W and BPC-fKL, which we interpret as the reverse KL suffering from the property that covers only one major mode, making the BPC-rKL posterior somewhat sub-optimal. In the 1 ipc (image per class) setting, BPC-W and BPC-fKL have comparable results, and as the pseudocoreset size increases, BPC-fKL is better if there is no augmentation in test time and BPC-W is better if not. It seems that the strength of BPC-W with higher performances with test time augmentations is due to the training method that exactly matches the pseudocoreset training trajectories and expert training trajectories that also be trained with the data augmentations. However, because the learning method of A-SGHMC is not accurate Bayesian learning, and it is known that data augmentation causes a cold posterior effect [30], the HMC results without augmentations are preferred.

To qualitatively examine the pseudocoresets trained with each divergence measure, we plot the learned pseudocoreset images in Fig. 1. We see that BPC-fKL looks the most smooth while BPC-rKL is the most noisy. The noisiness of the images learned by BPC-rKL may be due to the nature of the reverse KL divergence, which makes it difficult to escape the mode captured by the initial pseudocoreset images. In contrast, BPC-W and BPC-fKL seem to learn better pseudocoreset as desired to include high-level shapes semantic features representative of each class. As a more quantitative measure of image noisiness, we plot the log amplitude of the diagonal components of the 2D Fourier transform of pseudocoresets in Fig. 2: BPC-rKL has the most high-frequency noise, as reflected in the qualitative examples. This result is consistent with previous studies showing that CNNs are not robust to high-frequency noise [26, 25]. Please refer to Appendix D for all pseudocoreset images.

## 5.3 Computational Cost

We show that BPC-fKL, in addition to achieving high performance in terms of accuracy and NLL, requires substantially lower computational costs compared to other pseudocoreset training methods. We measure the computational costs of each method on CIFAR10 with fixed gradient steps $L_{\mathbf{u}} = 30$. We use 32 cores of Intel Xeon CPU Gold 5120 and 4 Tesla V100s. We note that BPC-W with many pseudocoresets per class cannot be trained on a single GPU due the memory limitations. Therefore, we used a parallel implementation using 4 GPUs in all the methods for a fair comparison. To compare the training time, we measure the time required for a single iteration by averaging across 100 iterations and repeat this process three times to select the largest value among them. Fig. 3a shows that the memory usage is significantly lower for BPC-fKL compared to BPC-W. This is because, unlike BPC-fKL, BPC-W must hold all gradient flows in memory for trajectory matching. As shown

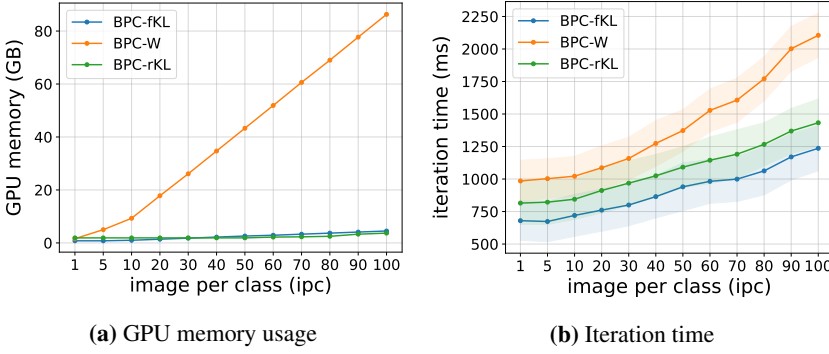

(a) GPU memory usage        (b) Iteration time

**Figure 3:** Comparison of computational costs of GPU memory and training time.

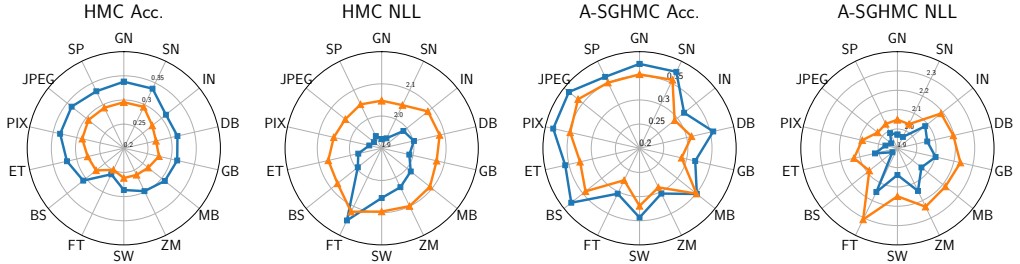

**Figure 4:** HMC and A-SGHMC results on CIFAR10-C. Each direction represents each type of corruption. Blue and orange represent BPC-fKL and BPC-W, respectively. All values are averaged over 30 runs with different random seeds. Acc → higher the better, NLL → lower the better.

in Fig. 3b, BPC-fKL also has the advantage of faster training time. Pseudocoreset training time increases roughly linearly with pseudocoreset size, but BPC-W's rate of increase is much faster than that of BPC-fKL. Combining the two results and Table 1, the pseudocoreset training using forward KL divergence is not only comparable in terms of accuracy and NLL but also remarkably efficient in both memory and training time, compared to BPC-W.

## 5.4 Robustness to Out-of-Distribution Inputs

Since one of the merits of the Bayesian approach is Bayesian model averaging which has been shown to improve robustness to distributional shift and calibration, we evaluate the performance of each pseudocoreset method on an out-of-distribution dataset CIFAR10-C [12]. Fig. 4 shows the accuracy and negative log-likelihood of each corrupted dataset for BPC-W and BPC-fKL. Note that we optimize each pseudocoreset with the clean CIFAR10 dataset and evaluate them on CIFAR10 with 14 corruptions. For A-SGHMC results, we apply differentiable augmentations to both the original dataset and coreset for training, but we do not use it for running HMC. As shown in Fig. 4, for both HMC and A-SGHMC, BPC-fKL achieves higher or comparable accuracy and lower negative log-likelihood than the BPC-W for all but one corruption, suggesting that the forward KL divergence is indeed an effective divergence measure for learning Bayesian pseudocoresets.

## 6 Conclusion

In this paper, we explored three divergence measures for Bayesian pseudocoresets: reverse KLD, Wasserstein distance, and forward KLD. We showed that existing dataset distillation methods can be linked to the Bayesian pseudocoresets with reverse KLD and Wasserstein distance, and further proposed a novel algorithm for learning Bayesian pseudocoresets by minimizing forward KL divergence. We empirically validated all three methods in terms of their ability to approximate the posterior distribution for real-world image datasets. Bayesian pseudocoresets with both Wasserstein distance and forward KL divergence can approximate the true posterior well and BPC-fKL is more effective in terms of computational cost and robustness to out-of-distribution data.

**Limitation** Despite showing promising results for the first time on Bayesian pseudocoresets for real datasets, there still exists a substantial performance gap between stochastic gradient MCMC on a pseudocoreset and the original dataset. Thus, a promising future direction is to analyze whether such pseudocoresets are useful for stochastic gradient MCMC algorithms when they are used together with mini-batches of the original dataset.

**Societal Impacts** Our work is hardly likely to bring any negative societal impacts. Nevertheless, we should be careful while learning pseudocoresets because any bias present in the original dataset can be transferred to the pseudocoresets. On a positive note, pseudocoresets can alleviate data privacy concerns by eliminating the need for access to the original dataset during downstream task learning.

## Acknowledgments

This work was partly supported by KAIST-NAVER Hypercreative AI Center, Korea Foundation for Advanced Studies (KFAS), Institute of Information & communications Technology Planning & Evaluation (IITP) grant funded by the Korea government (MSIT) (No.2019-0-00075, Artificial Intelligence Graduate School Program (KAIST), No. 2021-0-02068, Artificial Intelligence Innovation Hub, No.2022-0-00713), and National Research Foundation of Korea (NRF) funded by the Ministry of Education (NRF-2021M3E5D9025030).

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
