# A Proofs

**Proposition A.1.** *Let $q_{\mathbf{u}}(\theta)$ set as Eq. 12. Assume $\theta^{(t-1)}$ is close enough to local optima, so that we have $\|\theta^{(t-1)} - \theta^{(t)}\| \ll 1$. Then we have*

$$\nabla_{\mathbf{u}} D_{\mathrm{KL}}[\pi_{\mathbf{u}} \| \pi_{\mathbf{x}}] \approx -\eta \nabla_{\mathbf{u}}\left( \nabla_\theta \ell(\mathbf{x}, \theta^{(t-1)})^\top \nabla_\theta \ell(\mathbf{u}, \theta^{(t-1)}) \right). \tag{13}$$

*Proof.* For notational simplicity, let $\theta_0 = \theta^{(t-1)}$. We can reparameterize $\theta \sim q_{\mathbf{u}}$ as

$$\theta = \theta_0 - \eta \nabla_\theta \ell(\mathbf{u}, \theta_0) + \Sigma^{1/2}\varepsilon, \quad \varepsilon \sim \mathcal{N}(0, I), \tag{23}$$

Assume that $\eta$ and $\Sigma$ are chosen such that $\|\eta \nabla_\theta \ell(\mathbf{u}, \theta_0) - \Sigma^{1/2}\varepsilon\| \ll 1$. Then we have

$$
\begin{aligned}
\mathbb{E}_{\pi_{\mathbf{u}}}[\mathbb{1}_M^\top \mathbf{f}(\mathbf{u}, \theta)] &\approx \mathbb{E}_\varepsilon[\mathbb{1}_M^\top \mathbf{f}(\mathbf{u}, \theta_0 - \eta \nabla_\theta \ell(\mathbf{u}, \theta_0) + \Sigma^{1/2}\varepsilon)] \\
&\approx \mathbb{E}_\varepsilon\left[ \mathbb{1}_M^\top\left( \mathbf{f}(\mathbf{u}, \theta_0) + \nabla_\theta \mathbf{f}(\mathbf{u}, \theta_0)(-\eta \nabla_\theta \ell(\mathbf{u}, \theta_0) + \Sigma^{1/2}\varepsilon) \right) \right] \\
&= \mathbb{1}_M^\top\left( \mathbf{f}(\mathbf{u}, \theta_0) - \eta \nabla_\theta \mathbf{f}(\mathbf{u}, \theta_0) \nabla_\theta \ell(\mathbf{u}, \theta_0) \right).
\end{aligned}
\tag{24}
$$

Similarly,

$$\mathbb{E}_{\pi_{\mathbf{u}}}[\mathbb{1}_N^\top \mathbf{f}(\mathbf{x}, \theta)] \approx \mathbb{1}_N^\top\left( \mathbf{f}(\mathbf{x}, \theta_0) - \eta \nabla_\theta \mathbf{f}(\mathbf{x}, \theta_0) \nabla_\theta \ell(\mathbf{u}, \theta_0) \right). \tag{25}$$

Note also that

$$
\begin{aligned}
\nabla_{\mathbf{u}} \log Z(\mathbf{u}) &= \nabla_{\mathbf{u}} \log \int \exp(\mathbb{1}_M^\top \mathbf{f}(\mathbf{u}, \theta)) \pi_0(\mathrm{d}\theta) \\
&= \mathbb{E}_{\pi_{\mathbf{u}}}[\nabla_{\mathbf{u}}(\mathbb{1}_M^\top \mathbf{f}(\mathbf{u}, \theta))] \\
&\approx \mathbb{E}_{q_{\mathbf{u}}}[\nabla_{\mathbf{u}}(\mathbb{1}_M^\top \mathbf{f}(\mathbf{u}, \theta))] \\
&= \mathbb{E}_\varepsilon\left[ \nabla_{\mathbf{u}}\left( \mathbb{1}_M^\top \mathbf{f}(\mathbf{u}, \theta_0 - \eta \nabla_\theta \ell(\mathbf{u}, \theta_0) + \Sigma^{1/2}\varepsilon) \right) \right] \\
&\approx \mathbb{E}_\varepsilon\left[ \nabla_{\mathbf{u}}\left( \mathbb{1}_M^\top\left( \mathbf{f}(\mathbf{u}, \theta_0) + \nabla_\theta \mathbf{f}(\mathbf{u}, \theta_0)(-\eta \nabla_\theta \ell(\mathbf{u}, \theta_0) + \Sigma^{1/2}\varepsilon) \right) \right) \right] \\
&= \nabla_{\mathbf{u}}\left( \mathbb{1}_M^\top\left( \mathbf{f}(\mathbf{u}, \theta_0) - \eta \nabla_\theta \mathbf{f}(\mathbf{u}, \theta_0) \nabla_\theta \ell(\mathbf{u}, \theta_0) \right) \right).
\end{aligned}
\tag{26}
$$

Plugging this into the KL gradient, we get

$$
\begin{aligned}
\nabla_{\mathbf{u}} D_{\mathrm{KL}}[\pi_{\mathbf{u}} \| \pi_{\mathbf{x}}] &= -\nabla_{\mathbf{u}} \log Z(\mathbf{u}) + \nabla_{\mathbf{u}} \mathbb{E}_{\pi_{\mathbf{u}}}[\mathbb{1}_M^\top \mathbf{f}(\mathbf{u}, \theta)] - \nabla_{\mathbf{u}} \mathbb{E}_{\pi_{\mathbf{u}}}[\mathbb{1}_N^\top \mathbf{f}(\mathbf{x}, \theta)] \\
&\approx \eta \nabla_{\mathbf{u}}\left( \mathbb{1}_N^\top \nabla_\theta \mathbf{f}(\mathbf{x}, \theta_0) \nabla_\theta \ell(\mathbf{u}, \theta_0) \right) \\
&= -\eta \nabla_{\mathbf{u}}\left( \nabla_\theta \ell(\mathbf{x}, \theta_0)^\top \nabla_\theta \ell(\mathbf{u}, \theta_0) \right).
\end{aligned}
\tag{27}
$$

$\square$

# B Experimental Details

Code is available at https://github.com/balhaekim/BPC-Divergences.

## B.1 Hyperparameter settings

**Training** In Table 2, we enumerate the hyperparameters used for our results in Section 5. Since we use expert trajectories for all methods to train the Bayesian pseudocoresets, we refer to hyperparameters related to expert trajectories, such as the number of SGD steps or the maximum random starting points, described in [8]. We found that a slightly shorter expert training step is better for BPC-fKL, so we used an expert step 1 epoch shorter than BPC-W. Another important hyperparameter for BPC-fKL is the inner SGD learning rate $\eta$. For each setting, we used the best learning rate from a hyperparameter sweep over $\{0.01, 0.02, 0.03, 0.04\}$. All other hyperparameters are same for all methods.

**Table 2:** Hyperparameters used for our best-performing experiments.

| | | $K$ | $T^+$ | $L_{\mathbf{u}}$ | $L_{\mathbf{x}}$ | $\eta$ | $S$ | $\Sigma_{\mathbf{u}}^{1/2}$ | $\Sigma_{\mathbf{x}}^{1/2}$ | $B$ |
|---|---|---|---|---|---|---|---|---|---|---|
| 1 ipc | BPC-rKL | 5000 | 2 | 50 | - | 0.01 | 10 | 0.01 | - | 1000 |
| | BPC-W | 5000 | 2 | 50 | 2 | - | - | - | - | - |
| | BPC-fKL | 5000 | 2 | 50 | 1 | 0.01 | 30 | 0.01 | 0.01 | - |
| 10 ipc | BPC-rKL | 5000 | 20 | 30 | - | 0.03 | 10 | 0.01 | - | 1000 |
| | BPC-W | 5000 | 20 | 30 | 2 | - | - | - | - | - |
| | BPC-fKL | 5000 | 20 | 30 | 1 | 0.03 | 30 | 0.01 | 0.01 | - |
| 20 ipc | BPC-rKL | 5000 | 30 | 30 | - | 0.03 | 10 | 0.01 | - | 1000 |
| | BPC-W | 5000 | 30 | 30 | 2 | - | - | - | - | - |
| | BPC-fKL | 5000 | 30 | 30 | 1 | 0.03 | 30 | 0.01 | 0.01 | - |

---

**Algorithm 2** Hamiltonian Monte-Carlo Sampling (HMC)

---

**Require:** Number of iteration $N$, initial sample distribution scale $\sigma_\theta$, initial momentum distribution scale $\sigma_r$, number of leapfrog step $m$, step size $\varepsilon$,
**Require:** Potential energy function $U(\mathbf{u}, \theta) = -\mathbb{1}_M^\top \mathbf{f}(\mathbf{u}, \theta) + \lambda\|\theta\|_2^2$ with a dataset $\mathbf{u}$ and the weight decay factor $\lambda$.
    Initialize $\theta^{(1)} \sim \mathcal{N}(0, \sigma_\theta^2)$.
    **for** $t = 1, \ldots, N$ **do**
        Resample momentum $r^{(t)} \sim \mathcal{N}(0, \sigma_r^2)$.
        Set $(\theta_0, r_0) = (\theta^{(t)}, r^{(t)})$, $\theta^{(t+1)} = \theta^{(t)}$.
        $r_0 \leftarrow r_0 - \frac{\varepsilon}{2}\nabla U(\mathbf{u}, \theta_0)$
        **for** $i = 1, \ldots, m$ **do**
            $\theta_i \leftarrow \theta_{i-1} + \varepsilon r_{i-1}$
            $r_i \leftarrow r_{i-1} - \varepsilon\nabla U(\mathbf{u}, \theta_i)$
        **end for**
        $r_m \leftarrow r_{m-1} - \frac{\varepsilon}{2}\nabla U(\mathbf{u}, \theta_m)$
        $(\hat{\theta}, \hat{r}) = (\theta_m, r_m)$
        Metropolis-Hastings correction:
        $u \sim$ Uniform$(0, 1)$
        $\rho = e^{H(\hat{\theta}, \hat{r}) - H(\theta^{(t)}, r^{(t)})}$
        **if** $u < \min(1, \rho)$ **then**
            $\theta^{(t+1)} = \hat{\theta}$
        **end if**
    **end for**

---

**Evaluation** The evaluation methods we used are summarized in Algorithm 2 and Algorithm 3. We sampled the momentum from a normal distribution with scale $\sigma_r$ only for initialization. During leapfrog steps, we simulated the Hamiltonian dynamics as if it came from a standard Gaussian. As mentioned in the main text, the tendency did not significantly change depending on sampling hyperparameters. Since our focus is on providing a fair comparison between each Bayesian pseudocoreset method rather than raw performance, we used a single set of hyperparameters to generate all results. We summarize the hyperparameters used for our evaluations in Table 3.

**Table 3:** Hyperparameters used for evaluations.

| | | $N$ | $m$ | burn | $\sigma_\theta$ | $\sigma_r$ | $\varepsilon$ | $\lambda$ | $\alpha$ | $T$ |
|---|---|---|---|---|---|---|---|---|---|---|
| ipc 1 | HMC | 20 | 20 | 10 | 0.1 | 0.01 | 0.05 | 0.5 | - | - |
| | A-SGHMC | 20 | 5 | 10 | 0.1 | 0.1 | 0.03 | 1.0 | 0.1 | 0.01 |
| ipc 10 | HMC | 100 | 5 | 50 | 0.1 | 0.1 | 0.01 | 1.5 | - | - |
| | A-SGHMC | 100 | 5 | 50 | 0.1 | 0.1 | 0.01 | 1.5 | 0.1 | 0.01 |
| ipc 20 | HMC | 100 | 5 | 50 | 0.1 | 0.1 | 0.01 | 1.5 | - | - |
| | A-SGHMC | 100 | 5 | 50 | 0.1 | 0.1 | 0.01 | 1.0 | 0.1 | 0.01 |

**Algorithm 3** Altered Stochastic Gradient Hamiltonian Monte-Carlo Sampling (A-SGHMC)

---

**Require:** Number of iteration $N$, initial sample distribution scale $\sigma_\theta$, initial momentum distribution scale $\sigma_r$, number of leapfrog step $m$, step size $\varepsilon$, momentum decay factor $\alpha$, noise scale $T$.
**Require:** Potential energy function $U(\mathbf{u}, \theta) = -\mathbb{1}_M^\top \mathbf{f}(\mathbf{u}, \theta) + \lambda \|\theta\|_2^2$ with a dataset $\mathbf{u}$ and the weight decay factor $\lambda$.
**Require:** Differentiable augmentation function $\mathcal{A}$ used during the pseudocoreset training.
 Initialize $\theta^{(1)} \sim \mathcal{N}(0, \sigma_\theta^2)$.
 Initialize momentum $r^{(1)} \sim \mathcal{N}(0, \sigma_r^2)$.
 **for** $t = 1, \ldots, N$ **do**
  $(\theta_0, r_0) = (\theta^{(t)}, r^{(t)})$.
  **for** $i = 1, \ldots, m$ **do**
   $\theta_i \leftarrow \theta_{i-1} + \varepsilon r_{i-1}$
   $r_i \leftarrow (1 - \alpha) r_{i-1} - \varepsilon \nabla U(\mathcal{A}(\mathbf{u}), \theta_i) + \mathcal{N}(0, 2\alpha T)$
  **end for**
  $(\theta^{(t+1)}, r^{(t+1)}) = (\theta_m, r_m)$
 **end for**

---

**Table 4:** BPC-W vs BPC-W with diagonal covariances

|  |  | A-SGHMC | |
|---|---|---|---|
|  |  | Acc ($\uparrow$) | NLL ($\downarrow$) |
| ipc1 | BPC-W | $0.2934_{\pm 0.0121}$ | $2.1400_{\pm 0.0333}$ |
|  | BPC-W with d.c. | $\mathbf{0.2959}_{\pm 0.0108}$ | $\mathbf{2.1173}_{\pm 0.0289}$ |
| ipc10 | BPC-W | $\mathbf{0.4890}_{\pm 0.0172}$ | $\mathbf{1.6971}_{\pm 0.0392}$ |
|  | BPC-W with d.c. | $0.4848_{\pm 0.0113}$ | $1.7163_{\pm 0.0248}$ |

### B.2 Implementation details for BPC-rKL

To obtain a Bayesian pseudocoreset with reverse KL divergence by Algorithm 1 in [19], we need to sample from an approximated pseudocoreset posterior at each step through MCMC methods such as Langevin dynamics or HMC. To simply implement this, we also approximate the pseudocoreset posterior by a Gaussian distribution with the mean of the end point of SGD training trajectories like BPC-W or BPC-fKL. In initial experiments, we tried using SGD training trajectories starting from a random initial point or a point on the expert trajectory, but we found that using the expert trajectory achieves better performance. We provide a detailed description of the BPC-rKL algorithm in Algorithm 4.

## C  Additional Experiments

### C.1  Extending BPC-W to Gaussians with diagonal covariances

In Section 3.2, we approximated the pseudocoreset posterior and the original posterior to Gaussian distributions with the same covariances to obtain BPC-W. As an extension, we tried approximating the two distributions using Gaussian distributions with diagonal covariances. The A-SGHMC results for these pseudocoresets are in Table 4. We found that the results are comparable, and the additional expressivity of a diagonal covariance did not further increase performance. It seems to be because both posteriors would be much more complicated to approximate with Gaussians with diagonal covariances. While using a more complex distribution family might improve performance, we use Gaussian distributions with the same covariance in all dimensions to obtain BPC-W results throughout this paper.

### C.2  Gaussian approximation in BPC-fKL

In this experiment, we investigate the effect of the hyperparameters of the Gaussian approximation on the performance of BPC-fKL. Firstly, we explore the number of Gaussian samples $S$ and variances $\Sigma_u^{1/2}, \Sigma_x^{1/2}$ in Eq. 20. Fig. 5a shows the accuracy of HMC as the function of the number of samples. Even though the estimation becomes more accurate as the number of samples increases, in Fig. 5a, the number of samples does not significantly improve the performance of pseudocoresets. Thus, we

---

**Algorithm 4** Bayesian Pseudocoresets with Reverse KL

---

**Require:** Set of expert parameter trajectories $\{\tau\}$ trained with $\mathbf{x}$, each parameter trajectory saves parameters at the end of training epochs.

**Require:** Number of updates with the pseudocoreset $L_{\mathbf{u}}$, total training steps $K$, maximum start epoch $T^+$, the number of Gaussian samples $S$, variance $\Sigma_{\mathbf{u}}$, inner SGD learning rate $\eta$, minibatch size $B$, pseudocoresets learning rate $\gamma$.

**Require:** Differentiable augmentation function $\mathcal{A}$ (Optional).

Initialize the pseudocoreset $\mathbf{u}$ by randomly selecting a subset of size $M$ from $\mathbf{x}$.

**for** $k = 1, \ldots, K$ **do**

    Sample an expert trajectory $\tau = \{\theta_*^{(r)}\}_{r=0}^T$.

    Randomly choose an epoch to start $r \leq T^+$ and initialize $\theta_{\mathbf{u}}^{(0)} = \theta_*^{(r)}$.

    **for** $t = 1, \ldots, L_{\mathbf{u}}$ **do**

        Update the network parameter $\theta_{\mathbf{u}}^{(t)} \leftarrow \theta_{\mathbf{u}}^{(t-1)} + \eta \nabla \mathbb{1}_M^\top \mathbf{f}(\mathcal{A}(\mathbf{u}), \theta_{\mathbf{u}}^{(t-1)})$.

    **end for**

    Sample random Gaussian noises $\{\varepsilon_{\mathbf{u}}^{(s)}\}_{s=1}^S \overset{\text{i.i.d.}}{\sim} \mathcal{N}(0, I)$.

    Obtain a minibatch of $B$ datapoints from the original dataset $\{x_1, \ldots, x_B\} \subset \mathbf{x}$.

    **for** $s = 1, \ldots, S$ **do**

        $g_s \leftarrow \left( \mathbf{f}(\mathcal{A}(x_b), \theta_{\mathbf{u}}^{(L_{\mathbf{u}})} + \Sigma_{\mathbf{u}}^{1/2} \varepsilon_{\mathbf{u}}^{(s)}) - \frac{1}{S} \sum_{s'=1}^S \mathbf{f}(\mathcal{A}(x_b), \theta_{\mathbf{u}}^{(L_{\mathbf{u}})} + \Sigma_{\mathbf{u}}^{1/2} \varepsilon_{\mathbf{u}}^{(s')}) \right)_{b=1}^B \in \mathbb{R}^B$

        $\tilde{g}_s \leftarrow \left( \mathbf{f}(\mathcal{A}(u_m), \theta_{\mathbf{u}}^{(L_{\mathbf{u}})} + \Sigma_{\mathbf{u}}^{1/2} \varepsilon_{\mathbf{u}}^{(s)}) - \frac{1}{S} \sum_{s'=1}^S \mathbf{f}(\mathcal{A}(u_m), \theta_{\mathbf{u}}^{(L_{\mathbf{u}})} + \Sigma_{\mathbf{u}}^{1/2} \varepsilon_{\mathbf{u}}^{(s')}) \right)_{m=1}^M \in \mathbb{R}^M$

        **for** $m = 1, \ldots, M$ **do**

            $\tilde{h}_{m,s} \leftarrow \nabla_u \mathbf{f}(\mathcal{A}(u_m), \theta_{\mathbf{u}}^{(L_{\mathbf{u}})} + \Sigma_{\mathbf{u}}^{1/2} \varepsilon_{\mathbf{u}}^{(s)}) - \frac{1}{S} \sum_{s'=1}^S \nabla_u \mathbf{f}(\mathcal{A}(u_m), \theta_{\mathbf{u}}^{(L_{\mathbf{u}})} + \Sigma_{\mathbf{u}}^{1/2} \varepsilon_{\mathbf{u}}^{(s')})$.

        **end for**

    **end for**

    **for** $m = 1, \ldots, M$ **do**

        $\hat{\nabla}_{u_m} \leftarrow -\frac{1}{S} \sum_{s=1}^S \tilde{h}_{m,s} (\frac{1}{B} \mathbb{1}_B^\top g_s - \frac{1}{M} \mathbb{1}_M^\top \tilde{g}_s)$.

    **end for**

    **for** $m = 1, \ldots, M$ **do**

        $u_m \leftarrow u_m - \gamma \hat{\nabla}_{u_m}$.

    **end for**

**end for**

---

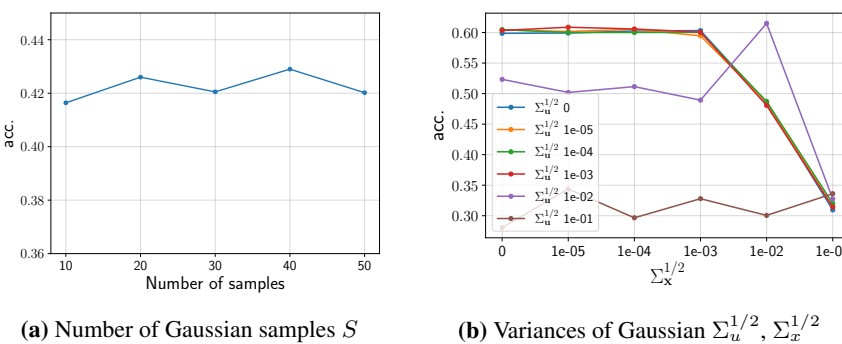

**(a)** Number of Gaussian samples $S$      **(b)** Variances of Gaussian $\Sigma_u^{1/2}, \Sigma_x^{1/2}$

**Figure 5:** Exploring the hyperparameters for Gaussian approximation in BPC-fKL. The pseudocoresets of size 10 images per class for CIFAR10.

use 30 samples for all the experiments. Fig. 5b, we show the accuracy with varying variances. The values of the x-axis are $\Sigma_{\mathbf{x}}^{1/2}$'s and $\Sigma_{\mathbf{u}}^{1/2}$'s are presented as colors. As the graph shows, too small variances are not much different from using the variance of 0, and when both values are 0.01 is the best and performance drops again for the variances larger than that. So we used both $\Sigma_{\mathbf{x}}^{1/2}$ and $\Sigma_{\mathbf{u}}^{1/2}$ of 0.01.

## C.3 Additional results on CIFAR10

Table 5 shows additional results for the CIFAR10 dataset when the pseudocoreset size is larger. Even in these cases, BPC-W and BPC-fKL effectively generate Bayesian pseudocoresets. Moreover, compared to Table 1, the 10-ipc pseudocoreset trained with BPC-fKL outperforms the 100-ipc random

**Table 5:** Performance of each Bayesian pseudocoreset method with {50, 100} images per class (ipc) on the CIFAR10 test dataset. We present results with HMC without augmentations during training. All values are averaged over ten random seeds.

| | | HMC | |
| --- | --- | --- | --- |
| | | Acc ($\uparrow$) | NLL ($\downarrow$) |
| ipc 50 | Random | $0.3922_{\pm 0.0037}$ | $2.1443_{\pm 0.0373}$ |
| | BPC-rKL | $0.3978_{\pm 0.0143}$ | $2.0692_{\pm 0.0803}$ |
| | BPC-W | $0.5424_{\pm 0.0092}$ | $\mathbf{1.4502}_{\pm 0.0720}$ |
| | BPC-fKL | $\mathbf{0.5557}_{\pm 0.0118}$ | $1.4619_{\pm 0.0504}$ |
| ipc 100 | Random | $0.4242_{\pm 0.0209}$ | $2.1430_{\pm 0.0703}$ |
| | BPC-rKL | $0.4220_{\pm 0.0200}$ | $2.1695_{\pm 0.1378}$ |
| | BPC-W | $\mathbf{0.5822}_{\pm 0.0494}$ | $1.6294_{\pm 0.1063}$ |
| | BPC-fKL | $0.5625_{\pm 0.0143}$ | $\mathbf{1.5841}_{\pm 0.0728}$ |

**Table 6:** HMC performances of the coresets and Bayesian psuedocoresets with 10 ipc on the CIFAR10 dataset. All values are averaged over ten random seeds.

| | Acc ($\uparrow$) | NLL ($\downarrow$) | ECE ($\downarrow$) | Brier score ($\downarrow$) |
| --- | --- | --- | --- | --- |
| Random | $0.2590_{\pm 0.0068}$ | $2.1820_{\pm 0.0241}$ | $0.1385_{\pm 0.0052}$ | $0.8595_{\pm 0.0063}$ |
| Herding | $0.3000_{\pm 0.0067}$ | $2.0343_{\pm 0.0189}$ | $0.1209_{\pm 0.0043}$ | $0.8231_{\pm 0.0055}$ |
| K-center | $0.1739_{\pm 0.0048}$ | $2.3934_{\pm 0.0132}$ | $0.1360_{\pm 0.0090}$ | $0.9125_{\pm 0.0032}$ |
| BPC-rKL | $0.3334_{\pm 0.0064}$ | $1.9516_{\pm 0.0178}$ | $\mathbf{0.1183}_{\pm 0.0038}$ | $0.7988_{\pm 0.0038}$ |
| BPC-W | $0.3538_{\pm 0.0111}$ | $1.9369_{\pm 0.0158}$ | $0.1457_{\pm 0.0110}$ | $0.8030_{\pm 0.0049}$ |
| **BPC-fKL** | $\mathbf{0.4361}_{\pm 0.0080}$ | $\mathbf{1.7198}_{\pm 0.0204}$ | $0.1538_{\pm 0.0049}$ | $\mathbf{0.7231}_{\pm 0.0049}$ |

coreset which has 10 times more images. The overall BPC-fKL results demonstrate that the forward KL divergence is effective for constructing the Bayesian pseudocoresets. For evaluations, we used same hyperparameters as the case of ipc 20 as described in Table 2 and Table 3, except that $\sigma_r$ is 0.01, $\varepsilon$ is 0.02 and $\lambda$ is 0.1.

**Other coreset baselines and evaluation metrics** We compared our results with other coreset baselines and other evaluation metrics for validating the quality of obtained posterior distributions more rigorously. We added Herding [29] and K-center [31] as another coreset baselines and for other evaluation metrics, we used expected calibration error (ECE) [21] and Brier score [2]. Herding constructs coresets by gathering samples close to the centers of the feature representations for each class and K-center constructs coresets by selecting multiple center points such that the distance between each data point is maximized while the distance between centers is minimized. To obtain the feature representations for both methods, we use a pre-trained ConvNet as in [13]. On the other hand, the ECE and Brier scores are conventional metrics for evaluating posterior qualities. Table 6 shows the HMC results of various metrics for coresets and psuedocoresets with 10 ipc. As [13] already shows that the coreset baselines underperform pseudocoresets for SGD, Table 6 also shows that pseudocoresets are better for Bayesian inference tasks through various metrics.

## C.4   Additional results on other datasets

In the main text, we trained Bayesian pseudocoresets only on the CIFAR10 dataset. To validate how well each method works on different datasets, we trained the pseudocoresets with a size of 1 image per class on other datasets, CIFAR100 and ImageNet. We can see that how well each method works when the number of classes is large with the CIFAR100 dataset and when the data dimension is large with the ImageNet dataset. The CIFAR100 dataset has the same image dimension as CIFAR10 but has 100 classes and ImageNet has $128 \times 128$ data dimension. For ImageNet, we use the same existing subset of the entire dataset, ImageNette, which consists of 10 classes and increased network architecture. Following previous work [8], we use a depth-5 ConvNet as the model architecture. As in the results on CIFAR10, Table 7 shows all three pseudocoresets are better than a random coreset. Moreover, BPC-W and BPC-fKL outperform BPC-rKL, demonstrating that the proposed divergence measures are effective.

**Table 7:** Performance of each Bayesian pseudocoreset method with 1 image per class (ipc) on the CIFAR100 and ImageNette test dataset. We present results with HMC without augmentations during training. All values are averaged over ten random seeds.

| | | HMC | |
|---|---|---|---|
| | | Acc (↑) | NLL (↓) |
| CIFAR100 | Random | $0.0420_{\pm 0.0025}$ | $4.7063_{\pm 0.0202}$ |
| | BPC-rKL | $0.0460_{\pm 0.0023}$ | $4.6665_{\pm 0.0293}$ |
| | BPC-W | $0.1035_{\pm 0.0053}$ | $\mathbf{4.2066}_{\pm 0.0180}$ |
| | BPC-fKL | $\mathbf{0.1055}_{\pm 0.0059}$ | $4.2366_{\pm 0.0220}$ |
| ImageNette | Random | $0.1572_{\pm 0.0264}$ | $2.4267_{\pm 0.0748}$ |
| | BPC-rKL | $0.2406_{\pm 0.0179}$ | $2.1896_{\pm 0.0222}$ |
| | BPC-W | $\mathbf{0.2876}_{\pm 0.0224}$ | $\mathbf{2.0977}_{\pm 0.0310}$ |
| | BPC-fKL | $0.2578_{\pm 0.0185}$ | $2.1520_{\pm 0.0325}$ |

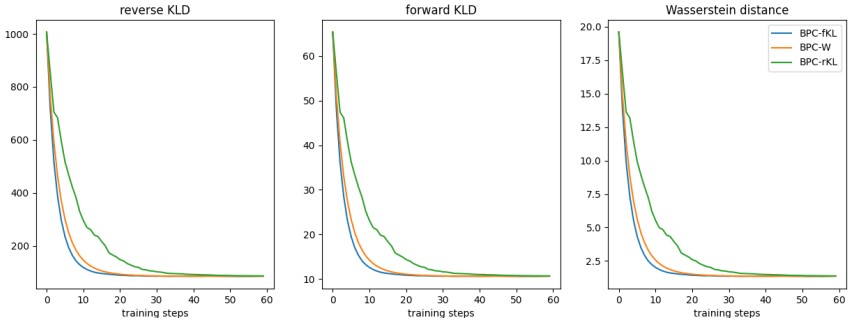

**Figure 6:** Divergence measures according to training steps where pseudocoreset size $M = 5$. Pseudocoresets trained with different divergence measures are represented by colors.

## C.5 Additional results on the synthetic dataset

To validate if the posterior distribution of each algorithm learns as intended, we trained pseudocoresets on the synthetic dataset of samples from 10-dimensional multivariate Gaussian distribution, whose posterior distribution is tractable. Given data samples $\{\mathbf{x}_1, \mathbf{x}_2, ..., \mathbf{x}_{100}\} \overset{i.i.d.}{\sim} \mathcal{N}(\theta, \Sigma)$, we trained pseudocoresets $\{\mathbf{u}_m\}_{m=1}^{M}$ of sizes $M = \{5, 20, 40, 60, 80, 100\}$ to have similar posterior distribution of $\theta \sim \mathcal{N}(\theta_0, \Sigma_0)$ with the true posterior which is tractable in this setting. Since we know that the exact posterior distribution is given by $\mathcal{N}(\theta_{\mathbf{u}}, \Sigma_{\mathbf{u}})$ where

$$\Sigma_{\mathbf{u}} = (\Sigma_0^{-1} + M\Sigma^{-1})^{-1}, \tag{28}$$

$$\theta_{\mathbf{u}} = \Sigma_{\mathbf{u}}(\Sigma_0^{-1}\theta_0 + \Sigma^{-1}\sum_{m=1}^{M} \mathbf{u}_m), \tag{29}$$

we validate each method by directly calculating the divergence measures between pseudocoresets posteriors and true posterior. Fig. 6 shows that all three methods work well in that all divergences are well reduced even when trained with the algorithm with different divergence measures in this simple synthetic setting. Also, as expected, Fig. 7 shows divergences decrease as pseudocoreset sizes increase.

## D Example images of each Bayesian pseudocoreset

Fig. 8 are the example images of CIFAR10 pseudocoresets of 10 images per class.

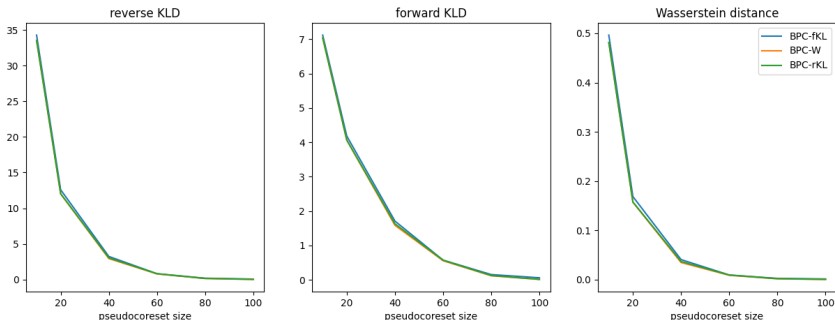

**Figure 7:** Divergence measures according to the pseudocoreset size. Pseudocoresets trained with different divergence measures are represented by colors.

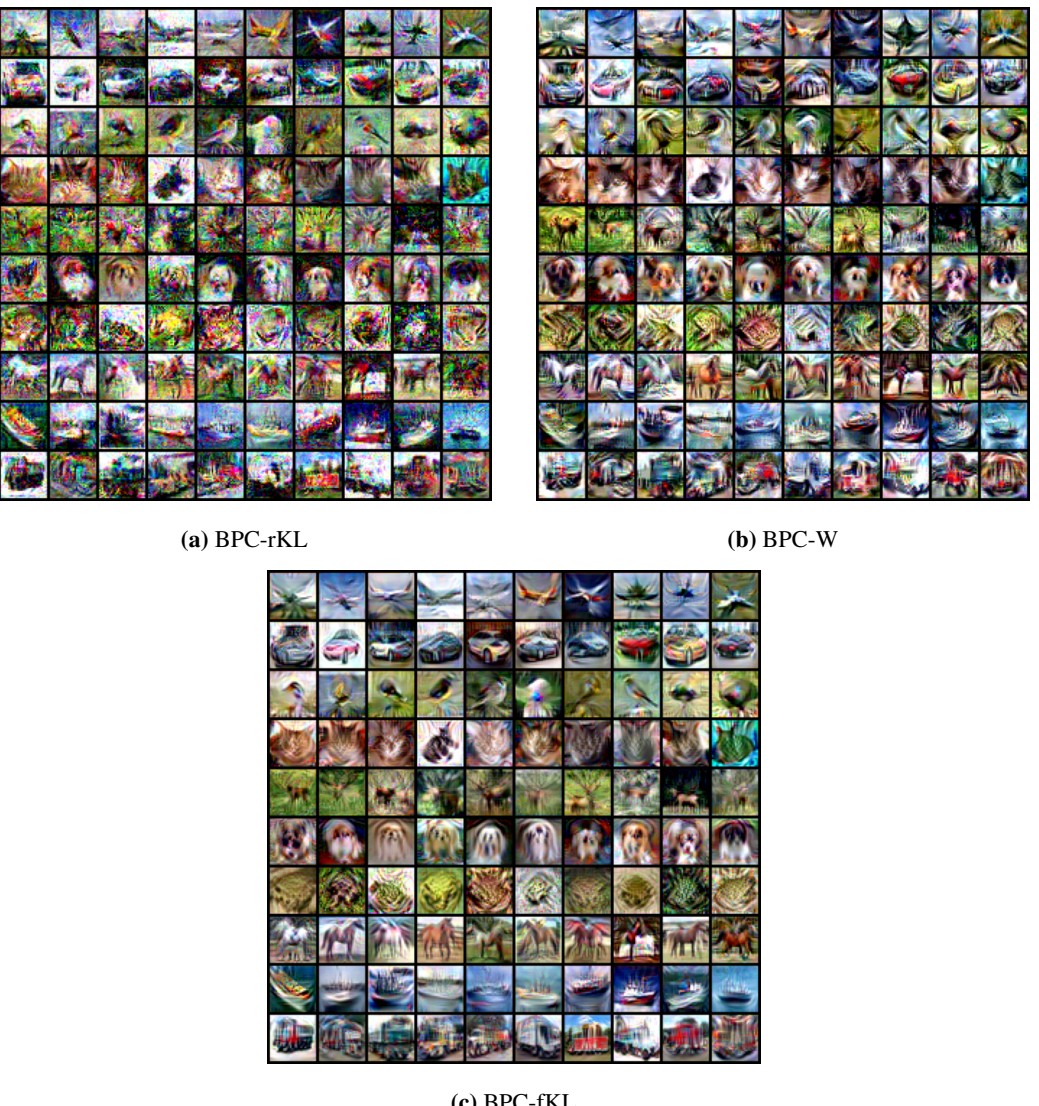

**(a)** BPC-rKL

**(b)** BPC-W

**(c)** BPC-fKL

**Figure 8:** Examples of Bayesian pseudocoresets. Each row is the pseudocoresets for each class.