# OpenReview forum: "On Divergence Measures for Bayesian Pseudocoresets"
_NeurIPS.cc/2022/Conference — NeurIPS 2022 Accept_

### Official Review · Reviewer_Pvtm · 2022-07-11

**Rating:** 7
**Confidence:** 4
**Soundness:** 3 good
**Presentation:** 3 good
**Contribution:** 3 good

**Summary:**

The paper studies learning Bayesian pseduocoresets, which is the task of a small set of "data" by conditioning on which the posterior is close to the posterior conditioned on the full dataset.
It establishes the framework in which two important ingredients are (1) the choice of divergence measure and (2) the choice of variational approximation.
Following this, it first recasts/reinterprets two existing, successful dataset distillation methods are Bayesian pseduocorsets learning with reverse KL/Wasserstein distance with Gaussian approximations to the posterior.
Then it proposes to use the forward KL which in theory doesn't suffer form the mode-seeking behaviour of reverse KL.
On a few real-world datasets, the proposed choice of forward KL is better than existing choice of divergence measures in terms of NLL and accuracy.

**Questions:**

- There are some discussions regarding contrastive divergence in section 3.3. I wonder if the divergence from [1] is related there or maybe a good choice of divergence in this framework.

[1] Ruiz, Francisco, and Michalis Titsias. "A contrastive divergence for combining variational inference and mcmc." In International Conference on Machine Learning, pp. 5537-5545. PMLR, 2019.

**Limitations:**

Limitations and potential negative societal impact are addressed.

**Strengths And Weaknesses:**

Pros
- The paper studies an important problem of scaling Bayesian methods to large datasets by learning Bayesian pseudocoresets. This approach can benefit a wide range of Bayesian inference methods.
- The paper establishes the framework of this problem, which is novel, and studies the relation to existing dataset distillation methods. The paper is significant as it (1) gives justification of some existing heuristics-based methods and (2) paves the way of further research in this direction.
- The paper is clearly written with good quality.
- Related works are discussed adequately.

Cons
- The experiments might be improved a bit.
  - I'd like to see at least one synthetic dataset where the posterior is tractable and see how the posterior each method learns differs from the true one.
  - Some classic Bayesian inference datasets could be used so that naive HMC results could be provided as a reference.

Writing
- Equation (1) is broken: I assume the $\pi_0$ in the middle shall be removed.

---

> ### Author Response · Authors · 2022-07-31
> **Response to Reviewer Pvtm**
>
> We sincerely appreciate your constructive comments. We respond to the individual comments below:
>
> **[Q1]** The experiments might be improved by showing a setting where the posterior is tractable.
> * We conducted an **additional experiment on the synthetic dataset of multivariate Gaussian distribution.** We considered a setting to infer the posterior distribution of the mean given observed samples. We first trained the pseudocoreset by each method we suggested, then validated them through exact divergences between two posteriors which are computable in this Gaussian setting.
> * As expected, all three methods work well in that all divergences are well reduced as the training progresses. Also, the larger the pseudocoreset size, the smaller the divergences. We have included the exact divergences between pseudocoreset posteriors and true posteriors both in terms of training steps and pseudocoreset sizes in **Appendix C.5.**
> ---
> **[Q2]** I wonder if the divergence from [1] is related to the contrastive divergence paragraph in section 3.3 or maybe a good choice of divergence in this framework.
> * Thanks for the pointer, we think that the variational contrastive divergence [1] can be a good candidate for the tractable divergence measures whose gradients can be computed efficiently during the pseudocoreset learning.
> ---
> **[Q3]** Typos.
> * Thank you for finding the typos and we have corrected them.
> ---
> ## References
> [1] Ruiz, Francisco, and Michalis Titsias. "A contrastive divergence for combining variational inference and mcmc." In International Conference on Machine Learning, pp. 5537-5545. PMLR, 2019.

---

> > ### Comment · Reviewer_Pvtm · 2022-08-08
> > **Reply to author response**
> >
> > Thanks for the extra experiment on synthetic data.
> > It looks like in the region of pseudocoreset size you explored, the divergences are far away from 0 but still are decreasing.
> > How much points do you need to get some number that is close to 0 or have the curves in figure 7 almost converged?

---

> > > ### Author Response · Authors · 2022-08-09
> > > **Response to the reply**
> > >
> > > Thank you for your response. We present additional results with larger pseudocoreset sizes, where we used larger learning rates (20.0) for all configurations for faster convergence.  For all three divergence measures, pseudocoresets of size larger than 60 could achieve divergence values less than 1, and pseudocoresets of size 100 could achieve near zero divergence values.
> > >
> > > **<BPC-fKL>**
> > > |             |  20   | 40   | 60   | 80    | 100   |
> > > |-------------|-------|------|------|-------|-------|
> > > | Reverse KL  | 12.16 | 3.10 | 0.89 | 0.18  | 0.09  |
> > > | Forward KL  | 4.09  | 1.65 | 0.62 | 0.15  | 0.09  |
> > > | Wasserstein | 0.16  | 0.03 | 0.01 | 0.002 | 0.001 |
> > >
> > > **<BPC-rKL>**
> > > |             |  20   | 40   | 60    | 80    | 100  |
> > > |-------------|-------|------|-------|-------|------|
> > > | Reverse KL  | 11.94 | 2.92 | 0.78  | 0.14  | 0.01 |
> > > | Forward KL  | 4.04  | 1.58 | 0.55  | 0.12  | 0.01 |
> > > | Wasserstein | 0.15  | 0.03 | 0.008 | 0.001 | 0.00 |
> > >
> > > **<BPC-W>**
> > > |             |  20   | 40   | 60    | 80    | 100  |
> > > |-------------|-------|------|-------|-------|------|
> > > | Reverse KL  | 11.86 | 2.90 | 0.77  | 0.13  | 0.01 |
> > > | Forward KL  | 4.03  | 1.57 | 0.55  | 0.11  | 0.01 |
> > > | Wasserstein | 0.15  | 0.03 | 0.008 | 0.001 | 0.00 | (edited)

---

> > > > ### Comment · Reviewer_Pvtm · 2022-08-09
> > > > **Reply to follow-up author response**
> > > >
> > > > Thanks for the new results.
> > > > You will need to include them to figure 7 to make the trend clear to readers
> > > > But the results looks great to me as they give an idea of how large the set is needed for a nearly perfect match.

---

> > > > > ### Author Response · Authors · 2022-08-09
> > > > > **Response to the reply**
> > > > >
> > > > > Thank you for your response. We have included them in Figure 7.

---

### Official Review · Reviewer_6N6Z · 2022-07-12

**Rating:** 7
**Confidence:** 4
**Soundness:** 3 good
**Presentation:** 3 good
**Contribution:** 4 excellent

**Summary:**

This paper reinterprets heuristic dataset distillation methods as approximate Bayesian pseudocoreset procedures, with different choices of divergence between the true posterior and the pseudocoreset approximation. Motivated by this analysis, they propose a new Bayesian coreset procedure based on an approximation to the forward KL divergence, and show empirically that it outperforms alternatives.

**Questions:**

Is there a general recipe for constructing Bayesian pseudocoresets using Gaussian approximations? In particular, can you enumerate the different choices of how the trajectory is managed, what divergence to pick, etc. Can you quantify the various sources of error in this approximation?

Small Suggestions:
Equation 1 has a typo, the comma in the middle (which makes it look as if the prior pi_0 is defined to be the posterior). Same typo appears in Eqn. 2

Equation 4: I believe u_m should not be bold, since it’s one datapoint, and the gradient on the RHS should be with respect to u_m not u, since the overall gradient should have dimensions the size of a single datapoint.

Line 67, the index should run to capital N not lowercase n. Might want to do the same in the contrastive divergence section, for consistent notation.

Line 89 - Typo: Traning -> Training

Line 180 - Check the grammar here and in the following lines (extends -> extend, etc.)

Line 193 - In fact, [16] and other Bayesian coreset methods demonstrate applications to real data (some real data is low dimensional, believe it or not).

Lines 215 - this paragraph is very unclear to me.


**Limitations:**

The authors are honest about some serious limitations of the method’s performance. I think it may be worth discussing privacy in the societal impacts section; pseudocoresets may potentially help increase data privacy while enabling downstream analysis on the one hand, while on the other hand their privacy properties have not be thoroughly studied

**Strengths And Weaknesses:**

This paper offers an insightful and unifying new view on heuristic dataset distillation techniques, which leads to improved practical methods. It is original (to my knowledge) and clearly written, and a significant contribution both to the dataset distillation and the Bayesian coresets literatures.
It suffers from two main weaknesses in terms of quality. First, the theoretical analysis is extremely rough, involving severe approximations (essentially treating the posterior as a Gaussian with zero variance, i.e. a delta function). There is a rich literature on Gaussian approximations to the posterior, including in particular Bernstein von Mises theorems. Moreover, SGD can be viewed as providing a Gaussian approximation to the  Bayesian posterior (Mandt, Hoffman, Blei, Stochastic Gradient Descent as Approximate Bayesian Inference). It would be valuable to establish more general and rigorous results in terms of Gaussian posterior approximations, and then derive the heuristic approximations as limiting cases when the covariance goes to zero (i.e. when the number of datapoints goes to infinity). The second major weakness is the empirical analysis. Despite motivating the work by Bayesian inference, there is not a rigorous evaluation of the quality of the posterior approximation (e.g. is the uncertainty correct), only evaluation of accuracy and log likelihood.

---

> ### Author Response · Authors · 2022-07-31
> **Response to Reviewer 6N6Z**
>
> We sincerely appreciate your constructive comments. We respond to the individual comments below:
>
> **[Q1]**  The theoretical analysis is extremely rough.
> * We agree that the current theoretical analysis is rough as you said (especially proposition 3.1), please note that our construction is to **reveal connections** between the dataset distillation and Bayesian pseudocoreset literatures through the lens of such simple variational approximations. Note also that we proposed a novel Bayesian pseudocoreset algorithm (BPC-fKL) that is comparable or sometimes better than BPC-W (MTT) with much less time and space complexity. As we proposed BPC-fKL in our framework, one can further propose more sophisticated variational approximations based on the rich theory of approximating posteriors with Gaussians as you suggested, or even considering alternative divergence measures.
> ---
> **[Q2]** There is not a rigorous evaluation of the quality of the posterior approximation.
> * We have added additional metrics to evaluate uncertainty quantification aspects of the methods, **including ECE and Brier scores. Please refer to the revision (Appendix C.3 and Table 6).**
>
>
> **<HMC performances of pseudocoresets>**
>
> |         |     ECE    | Brier score |
> |---------|:----------:|:-----------:|
> | Random  |   0.1385   |    0.8595   |
> | BPC-rKL | **0.1183** |    0.7988   |
> | BPC-W   |   0.1457   |    0.8030   |
> | BPC-fKL |   0.1538   |  **0.7231** |
>
> ---
>
> **[Q3]** Is there a general recipe for constructing Bayesian pseudocoresets using Gaussian approximations?
> * We thank you for your valuable question, which can potentially strengthen our paper. Generally, we find BPC-fKL to be comparable with BPC-W and better than BPC-rKL. Considering the heavy training cost of BPC-W due to the backpropagation through the unrolling step, BPC-fKL would be an attractive alternative to BPC-W when the **training resources are limited** or the target model has an extremely large number of parameters.
> * For the Gaussian approximations, we have tried constructing covariance matrices using the empirical statistics computed from the sample trajectories (e.g., running SGHMC, collecting the posterior samples, and computing empirical covariance matrices), but it did not result in significant improvement over the vanilla Gaussian approximation with constant covariances. Designing a flexible yet tractable variational approximation would be an important future research direction under our framework.
> ---
> **[Q4]** Can you quantify the various sources of error in this approximation?
> * Needless to say the approximation error of the Gaussian distribution, in practice, when we are constructing the full-data posterior, we use the mini-batch sampled from the entire dataset. The gradient noise due to the mini-batch sampling also contributes to the approximation error. Also, most of the divergence measures are intractable, so we rely on the stochastic gradients to minimize them. Although reparametrization tricks for the Gaussian sampling can reduce the variance of the stochastic gradient to some extent, there still is a gap between the true gradient of the divergence measures and the stochastic gradients.
> ---
> **[Q5]** I think it may be worth discussing privacy in the societal impacts section.
> * Thank you for your valuable comment. We have added the positive aspect of pseudocoresets that help increase data privacy.
> ---
> **[Q6]** Typos and unclear paragraphs.
> * Thank you for finding the typos and we have corrected them and rewrote the paragraphs more clearly.

---

### Official Review · Reviewer_RC9c · 2022-07-16

**Rating:** 5
**Confidence:** 4
**Soundness:** 3 good
**Presentation:** 3 good
**Contribution:** 2 fair

**Summary:**

The paper unifies the Bayesian Pseudocoresets and existing dataset distillation methods in certain settings. It investigates multiple choice divergence measurements and proposes an algorithm using forward KL-divergence to construct Bayesian Pseudocoresets.

**Questions:**

1. The assumption of proposition 3.1 should be defined clearer that $\Vert \theta_{t} - \theta_{t-1}\Vert$ should be sufficiently small.
2. Results shown in figure 1 is not very illustrative due to their size. Could you pick some representatives and move the rest to the appendix? Could you demonstrate the differences quantitively rather than qualitatively?
3. Could the performance gap between the using psuedocoreset and the entire dataset be indicated by the divergence?
4. Could you clarify the bias amplification on the downstream task mentioned in line 299? The Bayesian Pseudocoreset algorithm [Dionysis et al. 2020] claims weight w could potentially help with reducing the computation burden when nonzero entries are small. Is there any reference or empirical results substantiating the claim that m doesn't help with the performance?

**Limitations:**

The limitation is well discussed. The negative societal impact is not applied.

**Strengths And Weaknesses:**

- ***Strengths***:
1. The direction of unifying existing data condensation and Bayesian Pseudocoresets is well-motivated.
2. The effectiveness of the proposed method is well illustrated.


- ***Problems***:
1. Proposition 3.1 relies on two factors: (1) $\Vert \theta_{t} - \theta_{t-1}\Vert$ is sufficiently small so that the expansions in eq 24 and eq 25 hold; (2) the gaussian approximation. The second approximation might be commonly used in practice, yet this choice should be carefully elaborated on since it plays a key role throughout the paper. The first assumption actually limits the efficacy of the proposition to large t when \theta_t is close to local optima instead of the whole trajectory as in Data Condensation (eq 6). Then the connection between BPC-rKL and the existing Dataset Distillation method is not as exact as claimed.
2. In section 3.2, the discussed connection between BPC-W and MTT accredits mostly to the simplified setting using the gaussian approximation. As the distance of two Gaussians is minimized when the variance is shared and the distance of the mean is minimized.
3. Theoretical-wise, the paper mainly leverages the gaussian approximation to unify the Bayesian Pseudocorset with data distillation methods. The strong reliance on the gaussian approximations without adequate elaboration downgrades its contribution.
4. In section 5, since BPC-rKL and BPC-W are not exactly equivalent to existing methods, the dataset distillation methods and the original Bayesian Pseudocoresets algorithm are expected as baselines while missing. Especially when the implementations of these algorithms are available.

***Minor Issues***:
1. Line 422, I think it should be $\approx$ rather than =.
2. Line 419, the expectation seems to be over $\pi_x$ rather than $\pi_u$.

---

> ### Author Response · Authors · 2022-07-31
> **Response to Reviewer RC9c (2/2)**
>
> **[Q6]** Gaussian approximations should be carefully elaborated throughout the paper.
> * While we agree on this point, we believe that we can develop enhanced versions of the previous approaches (BPC-rKL, BPC-W) or the one we proposed (BKC-fKL) with more elaborated choices of the variational posteriors, which is a natural future research direction. Still, as noted by the reviewer 6N6Z, “There is a rich literature on Gaussian approximations to the posterior, including in particular Bernstein von Mises theorems”; that is, Gaussian approximation often makes sense with properly chosen covariance matrices. Ours is a more naive version of such approaches with the covariance matrices simply set as spherical matrices with constant variance values.
> ---
> **[Q7]** The relation between BPC-W and MTT accredits mostly to the simplified setting using the gaussian approximation.
> * As you said, our reinterpretation of MTT as BPC-W hinges on a simplest possible Gaussian approximation, we believe that this suggests a way of improving MTT by employing more sophisticated variational distributions, for instance advanced covariance matrices or even non-Gaussian variational posteriors. The key desiderata for those lines of work would be how to construct flexible yet tractable variational posteriors on the fly during the training.
> ---
> **[Q8]** In section 5, BPC-rKL and BPC-W are not exactly equivalent to existing methods.
> * **BPC-W is exactly equivalent to MTT**, and we used their code as a starting point for our implementation for the other methods. We used the name BPC-W for consistency throughout the paper, as our paper focuses on providing a unified view of pseudocoreset methods.
> * BPC-rKL corresponds to the Bayesian pseudocoreset [4] with the variational posteriors replaced with simple Gaussian from the Laplace approximation. This is mainly **due to the quadratic complexity of constructing Hessian matrices in the Laplace approximation** during the training in high-dimensional space.
> ---
> **[Q9]** Typos.
> * Thank you for finding the typos and we have corrected them.
> ---
> ## References
> [1] Lisa Anne Hendricks, Kaylee Burns, Kate Saenko, Trevor Darrell, and Anna Rohrbach. “Women also snowboard: Overcoming bias in captioning models.” In European Conference on Computer Vision, pages 793–811. Springer, 2018.
>
> [2] Pierre Stock and Moustapha Cisse. “Convnets and imagenet beyond accuracy: Understanding mistakes and uncovering biases.” In Proceedings of the European Conference on Computer Vision (ECCV), September 2018.
>
> [3] Tianlu Wang, Jieyu Zhao, Mark Yatskar, Kai-Wei Chang, and Vicente Ordonez. “Balanced datasets are not enough: Estimating and mitigating gender bias in deep image representations.” In Proceedings of the IEEE/CVF International Conference on Computer Vision (ICCV), October 2019.
>
> [4] D. Manousakas, Z. Xu, C. Mascolo, and T. Campbell. “Bayesian pseudocoresets.” In Advances in Neural Information Processing Systems 33 (NeurIPS 2020), 2020.
>
> [5] Shao, Rulin, et al. "On the adversarial robustness of vision transformers." arXiv preprint arXiv:2103.15670 (2021).
>
> [6] Park, Namuk, and Songkuk Kim. "Blurs behave like ensembles: Spatial smoothings to improve accuracy, uncertainty, and robustness." International Conference on Machine Learning. PMLR, 2022.

---

> > ### Comment · Reviewer_RC9c · 2022-08-09
> > **Reply to the Responses**
> >
> > I appreciate the detailed responses by the authors. I believe the significance of the work is more about its positioning in the literature and therefore I've updated my score. Here are my remaining concerns.
> >
> > - The paper relies on the reasonable Gaussian approximation. I expect the reasoning for the choice should be carefully elaborated in the revised paper. Also, the paper should make it clear that the equivalency between the Bayesian Pseudocoreset methods and data distillation methods does not necessarily rely on approximation. The approximation simplifies the analysis but probably is not the necessary condition for the intrinsic connection.
> > - I might misunderstand here. The eq (12) in proposition 3.1 does not match eq (6) of the DC method as the latter relies on the trajectory of $\theta$ while proposition 3.1 only holds for near optimum $\theta$. If so, the results on DC should be added for comparison.
> > - The equivalency between minimizing the divergence of the (sub)Gaussian distributions and minimizing the distance of empirical learning losses are typically expected. Even the analysis of proposition 1 doesn't provide significant insights into the connections.

---

> > > ### Author Response · Authors · 2022-08-09
> > > **Reply to the reply**
> > >
> > > We appreciate your effort to review our paper and responses.
> > >
> > > **[Q1]** The paper relies on the reasonable Gaussian approximation. I expect the reasoning for the choice should be carefully elaborated in the revised paper.  Also, the paper should make it clear that the equivalency between the Bayesian Pseudocoreset methods and data distillation methods does not necessarily rely on approximation. The approximation simplifies the analysis but probably is not the necessary condition for the intrinsic connection.
> > > - Thanks for your suggestion that can enhance the presentation of the paper. As you suggested, we added relevant discussions in the revision (line 127).
> > > ---
> > > **[Q2]**  The eq (12) in proposition 3.1 does not match eq (6) of the DC method as the latter relies on the trajectory of θ  while proposition 3.1 only holds for near optimum θ. If so, the results on DC should be added for comparison.
> > > - As you mentioned and we pointed out in line 127, the relation in proposition 3.1 is only for near optimum. So DC and BPC-rKL are not exactly equivalent, but BPC-rKL with Gaussian approximation reduces to DC when the learning trajectory of theta reaches local optimum. As you suggested, we've additionally conducted experiments with DC and present partial results here, and the rest will be put into the paper as soon as completed.
> > >
> > > **<Dataset Condensation>**, CIFAR-10, HMC
> > > |        |        Acc        |        NLL        |
> > > |--------|:-----------------:|:-----------------:|
> > > | ipc 1  | 0.2678$\pm$0.0090 | 2.2790$\pm$0.0573 |
> > > | ipc 10 | 0.3753$\pm$0.0131 | 1.8489$\pm$0.0211 | (edited)

---

> ### Author Response · Authors · 2022-07-31
> **Response to Reviewer RC9c (1/2)**
>
> We sincerely appreciate your constructive comments. We respond to the individual comments below:
>
> **[Q1]** The assumption of proposition 3.1 should be defined clearer that $‖θ_t−θ_{t−1}‖$ should be sufficiently small.
> * We have **updated the paper to be more explicitly clear about the assumption in proposition 3.1** about the magnitude of the parameter update and the fact that the approximation introduced here is applicable when close to convergence. Please also note that this proposition only compares BPC-rKL with DC, and our analysis of BPC-W and BPC-fKL does not rely on this approximation.
> ---
> **[Q2]** Results shown in figure 1 are not very illustrative due to their size. Could you pick some representatives and move the rest to the appendix? Could you demonstrate the differences quantitatively rather than qualitatively?
> * In the revised version, Fig 1 in the main text shows **fewer example images** to look better. We also added quantitative experiments on the learned images to supplement the qualitative observations about these images. **We took the Fourier transform of learned synthetic images and plotted the log amplitude vs. frequency in Fig 2.** BPC-rKL has the most high-frequency noise, substantiating our previous qualitative observation that “BPC-rKL is the most noisy”. Corresponding to the previous works [5,6] that discussed high frequency noises interfering with the training, the most noisy BPC-rKL performs the worst.
>
> ---
> **[Q3]** Could the performance gap between the using pseudocoreset and the entire dataset be indicated by the divergence?
> * Table 1 of the revised paper includes SGHMC performance on the **entire CIFAR10 dataset.** Using the same architecture, 3-layer ConvNet, we got an accuracy of 0.7383+0.0052 and an nll of 0.9387+0.0152.
> * To more directly verify whether the divergences we consider are a good measure of convergence to the true posterior, we added new experiments to Appendix C.4. We consider a **synthetic Gaussian dataset** where we can exactly compute the divergence with the true posterior. This experiment shows that the  divergences between the true posterior and the pseudocoreset posterior decrease as training progresses and converges to a low value.
> ---
> **[Q4]** Could you clarify the bias amplification on the downstream task mentioned in line 299?
> * Previous works [1,2,3] discuss the tendency of deep learning models to amplify unintended bias in datasets. Although their claim was not explicitly about pseudocoresets, we believe their reasoning also extends to pseudocoresets, which are similarly (synthetic) datasets. For clarity, we removed the sentence on amplification because it has not yet been studied exactly for the pseudocoresets.
> ---
> **[Q5]** Is there any reference or empirical results substantiating the claim that weight doesn't help with the performance?
> * We conducted experiments both with and without the learnable weights. However, since they were just similar or worse with weights in our setting, we simply tried to learn pseudocoresets only. These are some results of our experiments with weights.
>
> **<HMC performances of BPC-fKL with and without learnable weights>**
>
> |                   |          acc          |          nll          |
> |-------------------|:---------------------:|:---------------------:|
> | ipc=1      | 0.3354 $\pm$ 0.0066 | 2.0253 $\pm$ 0.0311 |
> | ipc=1  (+weights) |   0.2851 $\pm$ 0.0231   |   2.2860 $\pm$ 0.0636   |
> | ipc=10         | 0.4294 $\pm$ 0.0101 | 1.7292 $\pm$ 0.0248 |
> | ipc=10 (+weights) |   0.4209 $\pm$ 0.0107   |   1.7429 $\pm$ 0.0188   |
> | ipc=20            | 0.4910 $\pm$ 0.0088     | 1.6279 $\pm$ 0.0264     |
> | ipc=20 (+weights) | 0.4960 $\pm$ 0.0099 | 1.6164 $\pm$ 0.0258 |

---

### Official Review · Reviewer_knYM · 2022-07-18

**Rating:** 5
**Confidence:** 3
**Soundness:** 3 good
**Presentation:** 2 fair
**Contribution:** 3 good

**Summary:**

The authors discuss Bayesian pseudocoreset methods that uses reverse KL, KL and Wasserstein metrics special cases. They link these algorithms to data distillation as different posterior choices. Their main method is the Bayesian pseudocoreset algorithm that replaces reverse KL with a forward KL algorithm. They test these algorithms with Cifar-10 dataset with random coresets baseline.

**Questions:**

Have you compared your method with another baseline? At least another coreset algorithm a bit more intelligent than random, such as weighted summation or using input points distances?



**Limitations:**

Yes

**Strengths And Weaknesses:**

Strengths:

Focusing/unifying on the Bayesian coresets and distillation methods is a good direction and influence for the academia. The paper might have a good potential to be proceeded with more future works.

Weaknesses:

The baseline is a random coreset which is a very weak baseline. However given that they apply their algorithm on an image dataset, this might be acceptable, and to be improved as a future work.

The paper needs to be organized and motivated better. The authors might more clearly mention the relations and related works for data distillation and bayesian coresets method, then unify them. There are some optimality requirements for these relations, it could be organized better. It would be good to emphasize more on the reasons that they were able to make the Bayesian algorithm tractable with a divergence perspective.

The ideas and proofs are too straightforward and the experiments are too preliminary.

It would be good to do one more pass for proofreading.

---

> ### Author Response · Authors · 2022-07-31
> **Response to Reviewer knYM**
>
> We sincerely appreciate your constructive comments. We respond to the individual comments below:
>
> **[Q1]** Have you compared your method with another baseline? At least another coreset algorithm a bit more intelligent than random, such as weighted summation or using input points distances?
>
> * In the revised submission, **we added two more sophisticated coreset baselines:  Herding [1] and K-center [2] (Appendix C.3, Table 6).** Herding gathers samples near centers of feature representations for each class and K-center selects multiple center points such that the distance between each data point is maximized while the distance between centers is minimized. We use a pre-trained ConvNet to obtain the feature representations for both methods. Similarly to how [3] reports that these baselines underperform pseudocoresets for SGD, our results also show that **pseudocoresets are better for Bayesian inference tasks.** Please refer to the revised paper for more results.
>
>
> **<HMC performances of coresets and pseudocoresets>**
>
> |          |         acc         |         nll         |
> |----------|:-------------------:|:-------------------:|
> | Herding  | 0.3000 $\pm$ 0.0067 | 2.0343 $\pm$ 0.0189 |
> | K-center | 0.1739 $\pm$ 0.0048 | 2.3934 $\pm$ 0.0132 |
> | BPC-rKL | 0.3334 $\pm$ 0.0064 | 1.9516 $\pm$ 0.0178 |
> | BPC-W | 0.3538 $\pm$ 0.0111 | 1.9369 $\pm$ 0.0158 |
> | **BPC-fKL** | **0.4361 $\pm$ 0.0080** | **1.7198 $\pm$ 0.0204** |
>
> ---
> **[Q2]** The relationship between dataset distillation and Bayesian pseudocoroesets needs to be more organized. It would be good to emphasize more on the reasons that they were able to make the Bayesian algorithm tractable with a divergence perspective.
>
> * Thanks for your valuable comment. As you suggested, **we revised the section in the introduction so that it can better highlight the connection** between dataset distillation and Bayesian pseudocoresets, and how the Bayesian pseudocoresets can be made scalable thanks to this connection.
> ---
> **[Q3]** The ideas and proofs are too straightforward and the experiments are too preliminary.
> * Although we agree that the proof of Proposition 3.1 is quite straightforward, **our core contribution is in revealing connections** between existing dataset distillation and Bayesian pseudocoreset methods, not the proposition or theory themselves. Based on our findings, one can further develop both dataset distillation or Bayesian pseudocoresets, as we did in our paper by minimizing alternative divergence measures and adopting advances in  dataset distillation to make Bayesian pseudocoreset methods more scalable.
> * To the best of our knowledge, our work is **the first to learn Bayesian pseudocoresets for real-world image classification datasets** with Bayesian neural networks, targeting SGHMC rather than vanilla SGD. We believe the experiments in the paper demonstrate the scalability of Bayesian pseudocoreset methods given our additional tricks, and we leave evaluation on ImageNet-scale data to future work.
> ---
> ## References
> [1] Max Welling. “Herding dynamical weights to learn.” In Proceedings of the 26th Annual International Conference on Machine Learning, pp. 1121–1128. ACM, 2009.
>
> [2] G W Wolf. “Facility location: concepts, models, algorithms and case studies.” 2011.
>
> [3] B. Zhao, K. R. Mopuri, and H. Bilen. “Dataset condensation with gradient matching.” In International Conference on Learning Representations, 2021.

---

### Author Response · Authors · 2022-07-31
**Summary of the Revision**

We really appreciate all the reviewers for their constructive comments. Here is the summary of the revision. The edited parts are marked as blue in the file.

* **More coreset baselines (Herding, K-center) and more uncertainty quantification metrics (ECE, Brier scores)** in Appendix C.3 as suggested by reviewers knYM, 6N6Z.
* Experiments with **synthetic data** where the exact divergence values are tractable to compute in Appendix C.5 as suggested by reviewer Pvtm.
* **Quantitative evaluation** of learned pseudocoreset images in Figure 2 as suggested by reviewer RC9c.
* Assumptions are made **more explicit in Proposition 3.1** as suggested by reviewer RC9c.
* Rewrote unclear paragraphs (especially in the part where we are motivating our work in the introduction) with more explanations, corrected typos pointed out by the reviewers.
* Moved section 5.3 to Appendix C.2 due to the page limit.

---

### Author Response · Authors · 2022-08-07
**A Gentle Reminder**

Dear reviewers,

We sincerely appreciate your efforts in reviewing our paper, and your constructive comments. We have responded to your comments, faithfully reflected them in the revision, and provided additional experimental results that you have requested. Could you please go over our responses and the revision since end of the final discussion phase is approaching? Please let us know there is anything else we need to clarify or provide.

Thanks, authors.

---

### Meta-Review · Area_Chair_KSXH · 2022-08-21

**Recommendation:** Accept
**Confidence:** Certain

**Metareview:**

There was a consensus among reviewers that this paper should be accepted. The authors formulate dataset distillation methods as approximate Bayesian pseudocoreset procedures which appears to be a novel viewpoint. They further propose a new coreset procedure and show that it performs well. The paper appears to be well-written.

**Award:**

No

---

### Decision · Program_Chairs · 2022-09-14

Accept